# Toolbox of FRET-based c-di-GMP biosensors and its FRET-To-Sort application for genome-wide mapping of c-di-GMP regulation

Liyun Wang [1], Gabriele Malengo[1], Ananda Sanches-Medeiros [1], Xuanlin Chen[1], Julian Pietsch[1], Nataliya Teteneva [1], Silvia González Sierra[1], Ming C. Hammond[2,3] & Victor Sourjik [1] ✉

C-di-GMP is a widespread second messenger that coordinates transitions between different lifestyles in bacteria. Levels of c-di-GMP are controlled by complex regulatory networks, and they can vary dynamically over a wide range of concentrations. To enable studies of c-di-GMP regulation under a variety of conditions, here we construct and characterize a large set of FRET-based c-di-GMP biosensors that undergo large FRET signal changes and display a stepwise coverage of diverse binding affinities, thus capable of sensitively detecting diverse cellular c-di-GMP concentrations. We subsequently apply different-affinity FRET biosensors from this toolbox to systematically investigate genome-wide network of c-di-GMP regulation in planktonic *Escherichia coli* cells by establishing FRET-To-Sort, which relies on FRET-based cell sorting of a barcoded transposon library. We observe prominent enrichment of mutations in two classes of flagellar genes among those affecting c-di-GMP levels, and demonstrate that inhibited flagellar rotation reduces biosynthesis of c-di-GMP due to increased proton motive force.

The second messenger cyclic diguanosine monophosphate (c-di-GMP), now recognized as a ubiquitous signaling molecule in bacteria, regulates a multitude of cellular processes that are primarily related to lifestyle transitions, including switch between motility and sessility, virulence, biofilm formation and cell cycle control[1,2]. In most bacterial species, c-di-GMP levels are controlled by a highly complex network of multiple diguanylate cyclases (DGCs) and phosphodiesterases (PDEs), which are responsible for the synthesis and the hydrolysis of c-di-GMP, respectively. These enzymes can respond to environmental and cellular cues and mediate cellular signaling via c-di-GMP binding effectors, including transcription factors[3,4] and post-translational regulators[5–7]. A large class of the latter are proteins containing PilZ domains, which are activated by c-di-GMP and generally function via protein-protein interactions, exemplified by the flagellar brake protein YcgR in *Escherichia coli*[6,7].

Monitoring c-di-GMP dynamics in vivo is critical to understand bacterial cell physiology. For that purpose, several c-di-GMP biosensors have been developed and employed over the last decade. These include transcriptional biosensors[8,9], in which a c-di-GMP-responsive promoter controls expression of a fluorescent protein gene. While being powerful in visualizing changes in c-di-GMP over generations, transcriptional reporters provide only an indirect readout and do not allow monitoring real-time dynamics. Hence, several protein-based biosensors for c-di-GMP with direct, real-time readouts have been developed. These include ratiometric biosensors that exploit c-di-GMP-induced dimerization of BldD to control fluorescence of fused fluorescent protein[10,11], and several biosensors based on Förster resonance energy transfer (FRET) that utilize c-di-GMP-binding PilZ-domain proteins fused to FRET pairs composed by donor and acceptor fluorescent proteins[12,13].

[1]Max Planck Institute for Terrestrial Microbiology & Center for Synthetic Microbiology (SYNMIKRO), Marburg, Germany. [2]Department of Chemistry and Henry Eyring Center for Cell & Genome Science, University of Utah, Salt Lake City, UT, USA. [3]Department of Chemistry, University of California, Berkeley, CA, USA. ✉e-mail: victor.sourjik@mpi-marburg.mpg.de

While the above-mentioned real-time biosensors have proven to be highly useful in monitoring c-di-GMP levels in bacteria, the number of available biosensors remains limited and each individual biosensor can only sensitively detect a narrow, typically one-decade range of c-di-GMP concentrations. Moreover, all previously developed biosensors exhibit relatively low binding affinities, limiting the range of detected c-di-GMP concentrations to those above ~200 nM. However, intracellular c-di-GMP concentrations are frequently lower[14–16], and they vary strongly among bacterial strains[14,17], between sessile and motile lifestyles[2,18,19], during the cell cycle[12,18,20], as well as upon changes of environmental factors[21], such as nutrients[22,23], oxygen[23] and antibiotics[24]. This full spectrum of physiologically relevant c-di-GMP concentrations cannot be currently well covered by the existing biosensors. Furthermore, direct comparison of results obtained from different existing biosensors is complicated by differences in their design. Therefore, capturing and comparing c-di-GMP dynamics across different conditions requires multiple biosensors with a consistent readout while spanning complementary and gradually differing affinities, including high-affinity biosensors that were not previously available.

We thus aim to develop a comprehensive toolbox of c-di-GMP biosensors with a uniform design but diverse affinities that cover the entire range of cellular c-di-GMP concentrations. For that, we make use of a library of 90 PilZ-domain proteins, homologous to *E. coli* YcgR, which were previously established for in-vitro c-di-GMP measurement using bioluminescence resonance energy transfer (BRET)[25], and convert it into in vivo FRET biosensors. To enable a high-throughput characterization of this library, we establish a flow cytometry-based protocol to robustly quantify FRET efficiency and to determine biosensor affinities. From this screening, we select a total of sixteen c-di-GMP FRET biosensors exhibiting large magnitude of response and covering diverse affinities, ranging from ~9 nM to ~1 μM, for the FRET biosensor toolbox.

Leveraging our toolbox, we establish an application, FRET-To-Sort, for genome-wide mapping of the c-di-GMP regulatory network in *E. coli*. FRET-To-Sort adapts cytometry-based quantification of FRET for fluorescence-activated cell sorting (FACS) of a barcoded Tn5 transposon library[26], enabling high-throughput identification of mutants with perturbed c-di-GMP levels, which should be widely applicable for mapping of c-di-GMP regulatory networks across various bacteria. We uncover complexity in the c-di-GMP regulatory network of *E. coli*, including a dual function of flagella in c-di-GMP regulation, with two different classes of flagellar mutations having opposite impacts on c-di-GMP levels. Specifically, class II flagellar gene mutations elevate c-di-GMP by suppressing transcription of the major *E. coli* PDE, whereas class III gene mutations impairing flagellar rotation reduce c-di-GMP via increased proton motive force (PMF), apparently sensed by the major *E. coli* DGC. Our results suggest that the latter mechanism might allow motile *E. coli* cells to perceive mechanical properties of their environment through their impact on flagellar rotation.

## Results

### Construction of the FRET-based c-di-GMP biosensor library

Upon c-di-GMP binding, PilZ-domain proteins are known to undergo a conformational change leading to an altered distance between their termini[5,27], which has enabled their previous use as FRET biosensors[12,13,22]. Such FRET biosensors can exhibit highly sensitive and real-time response to dynamic changes in c-di-GMP levels, but the detection range of individual biosensors is limited to roughly one decade, centered around its dissociation constant ($K_D$). We therefore constructed a toolbox of PilZ-domain-based FRET biosensors with the uniform design but covering diverse affinities, which offers a versatile choice of biosensors to investigate c-di-GMP levels under different conditions, including high-affinity biosensors for low concentrations. For that, we used the previously assembled library of 90 YcgR

homologues from different organisms[25]. These proteins were then fused to mNeonGreen (mNG, FRET acceptor) and mTurquoise2 (mTq2, FRET donor) fluorescent proteins (Fig. 1a and Supplementary Data 1). The resulting biosensors were named according to their positions in the library.

To evaluate their responsiveness in vivo, these putative c-di-GMP biosensors were expressing in two *E. coli* strains deleted for either the major DGC (Δ*dgcE*) or the major PDE (Δ*pdeH*)[2,28], and producing either very low (Δ*dgcE*) or very high (Δ*pdeH*) intracellular c-di-GMP levels, respectively. Differences in FRET between these two backgrounds thus reflect signal changes upon c-di-GMP binding by each biosensor (Fig. 1a). To enable an efficient high-throughput characterization of the biosensor library, we established a flow-cytometry based measurement of FRET in *E. coli* that relies on the sensitized acceptor emission[29]. The ratio of signals in the FRET channel (sensitized acceptor emission) and the donor emission channel (FRET ratio; Supplementary Fig. 1a,b) differed significantly between biosensors expressed in Δ*pdeH* and Δ*dgcE* strains grown in moderately rich tryptone broth (TB) medium until post-exponential phase ($OD_{600}$ ~ 0.6) (Fig. 1b–d), although the distributions across the population were broad. Notably, while some biosensors (e.g., D3; Fig. 1b,d) showed elevated FRET at higher c-di-GMP levels (in Δ*pdeH* background), others exhibited reduced FRET (e.g., C1; Fig. 1c,d). For the latter biosensor, which is based on the same YcgR (STM 1798) from *Salmonella* as the previously published FRET biosensor, our result is consistent with the previous analysis[12,13,22]. This indicates that different YcgR homologues undergo opposite conformational changes upon binding c-di-GMP (Fig. 1a).

Although being a useful proxy, the FRET ratio provides only an imperfect readout of FRET efficiency, as it does not take into account the direct excitation of acceptor and the bleed-through from the donor emission into the FRET channel[29] (Supplementary Fig. 1a). These could make comparisons of the absolute values of FRET ratio between different biosensors and experiments less precise, particularly in cases where biosensor expression levels differ. We thus corrected for these factors, by adapting the previously published method[30] to calculate FRET efficiency from flow cytometry data more accurately (Methods and Supplementary Fig. 1b–d). This approach revealed larger changes in FRET efficiency (Fig. 1e) between high and low c-di-GMP levels compared to those observed using the FRET ratio (Fig. 1d). We further validated our approach by comparing FRET efficiency values obtained by flow cytometry to their direct measurement using acceptor photobleaching microscopy[6,31,32], an accurate but lower-throughput method of FRET efficiency quantification (Supplementary Fig. 1e,f). For 46 randomly chosen biosensors, expressed in the Δ*dgcE* strain, the two approaches showed an excellent quantitative agreement (Fig. 1f). Furthermore, values of FRET efficiency were confirmed to be insensitive to the biosensor expression levels, in contrast to the FRET ratios (Supplementary Fig. 1g,h), which is important for reliable comparison between c-di-GMP at different growth conditions and in different strains, and for comparisons of different biosensors.

### A toolbox of c-di-GMP FRET biosensors with diverse affinities

Using the above established high-throughput assay, we proceeded to compare FRET efficiencies under high and low c-di-GMP backgrounds for the entire biosensor library. Most biosensors displayed high FRET efficiencies in both strains, with a median value of ~22% (Supplementary Fig. 2), which is consistent with the expected proximity of the donor and acceptor fluorophores within the fusion protein. Biosensors that showed low values of FRET efficiency (<10%) and/or high variability between replicates (Supplementary Fig. 2 and Supplementary Fig. 3) were excluded from further analysis. Many of the remaining biosensors showed significant differences in FRET efficiencies between the Δ*pdeH* and the Δ*dgcE* strains (Fig. 2a and Supplementary Fig. 3), confirming that their conformation changed upon c-di-GMP binding. A similar number of biosensors showed either increased or decreased

FRET at higher c-di-GMP levels, corresponding to two possible conformational changes induced by c-di-GMP (Fig. 1a).

For further characterization, we chose sixteen biosensors (Table 1 and Supplementary Table 1) with FRET efficiency above 15% in both $\Delta pdeH$ and $\Delta dgcE$ strains, and with most pronounced difference between these strains, indicating large conformational change dependent on cellular c-di-GMP levels (orange symbols in Fig. 2a). In order to determine the affinity of these selected biosensors to c-di-GMP in a high-throughput manner, we rendered biosensor-expressing $\Delta dgcE$ cells permeable to extracellular c-di-GMP by treating them with toluene[33]. This approach was previously used for FRET biosensor characterization, yielding affinities similar to those measured for purified proteins[34]. All sixteen biosensors exhibited reproducible dose-dependent changes in FRET when incubated with different concentrations of c-di-GMP (Fig. 2b and Supplementary Fig. 4a). The sign of their response – either increase or decrease in FRET upon c-di-GMP binding – was consistent with the in-vivo difference between $\Delta pdeH$ and $\Delta dgcE$ strains (Fig. 2a and Supplementary Fig. 3). The measured affinities of the biosensors to c-di-GMP stepwise covered the range from ~9 nM to ~1 μM, encompassing biosensors with low ($K_D$ ~ 200–1000 nM), medium ($K_D$ ~ 50–200 nM) and high ($K_D$ ~ 9–20 nM) affinity (Table 1). Particularly, the high-affinity

biosensors enable measurements of low c-di-GMP concentrations that are physiologically relevant for *E. coli* (Supplementary Fig. 4b) and other bacteria but were not accessible to existing FRET biosensors[12,13]. Measurements of $K_D$ values in permeabilized cells were further validated by assessing c-di-GMP induced changes in the fluorescence emission spectra of five selected purified biosensors in vitro (Supplementary Fig. 5a,b). The resulting in vitro-determined values correlated well with measurements in permeabilized cells (Supplementary Fig. 5c,d and Table 1). Our $K_D$ measurements also showed good agreement with previously reported values for several YcgR homologues[12,25,35,36] (Table 1), with moderate discrepancies (up to ~2-fold) likely arising from differences in methods, buffer compositions, and/or the effect of fluorescent protein fusions on binding affinity.

To further confirm that the biosensor affinities in permeabilized and intact cells are consistent, we next compared FRET values in intact wildtype, $\Delta pdeH$ and $\Delta dgcE$ cells. We reasoned that a greater similarity between FRET values in the wildtype (intermediate c-di-GMP) and $\Delta pdeH$ strain (high c-di-GMP) must reflect higher relative occupancy of the biosensor in wildtype cells, and thus higher biosensor affinity for c-di-GMP. Indeed, there was a clear correlation between the biosensor affinity determined in permeabilized cells and such relative in-vivo occupancy (Fig. 2c and Supplementary Fig. 6a, b). Notably, our

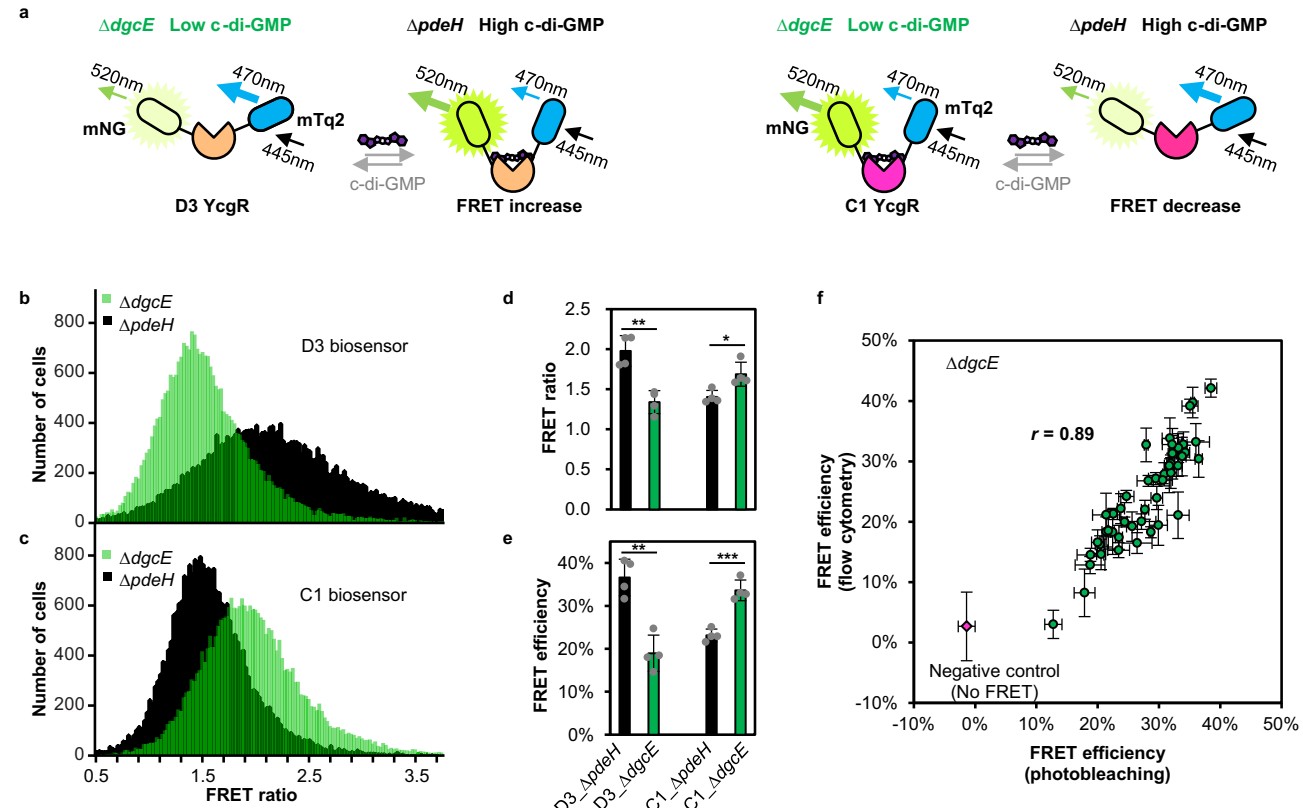

**Fig. 1 | Construction and characterization of c-di-GMP FRET biosensor library.** **a** Schematic representation of the FRET-based c-di-GMP biosensors. To screen potential biosensors for their sensitivity to changes in c-di-GMP, fusion proteins were expressed in $\Delta pdeH$ and $\Delta dgcE$ strains, which respectively have high or low levels of c-di-GMP, as indicated. Upon binding c-di-GMP, conformational change in the biosensor might either decrease or increase the distance between the donor, mTurquoise2 (mTq2), and acceptor, mNeonGreen (mNG), fluorophores, resulting in respectively increased (left) or decreased (right) energy transfer from mTq2 to mNG. Donor excitation (445 nm) and donor and acceptor emission (470 nm and 520 nm, respectively) are mean emission wavelength used in flow cytometry measurements. Proximity-dependent energy transfer reduces donor emission and increases acceptor emission. Histograms of the FRET ratio (the ratio of fluorescence intensity in the FRET channel over that in the donor channel upon donor

excitation) values in the population of either $\Delta pdeH$ or $\Delta dgcE$ cells expressing D3 (**b**) or C1 (**c**) biosensors. Values of the FRET ratio (**d**) and the FRET efficiency (**e**) for D3 and C1 biosensors in $\Delta pdeH$ and $\Delta dgcE$ strains, as indicated, calculated using median fluorescence intensities recorded in each flow cytometry channel (illustrated in Supplementary Fig. 1b-d). *$P < 0.05$; **$P < 0.01$; ***$P < 0.001$. $P$ values were calculated using unpaired two-tailed *t-test*, $n = 4$ biological replicates. Data are presented as mean ± SD. Exact $P$ values are reported in the Source Data file. **f** Comparison of the values of FRET efficiency calculated using calibrated flow cytometry with those measured using acceptor photobleaching microscopy for different potential biosensors in the $\Delta dgcE$ strain. The negative control corresponds to mTq2 and mNG co-expressed from different plasmids in the $\Delta dgcE$ strain. $n = 3$ biological replicates. Data are presented as mean ± SD. Pearson correlation coefficient $r$ is shown. Source data for (**b**–**f**) are provided as a Source Data file.

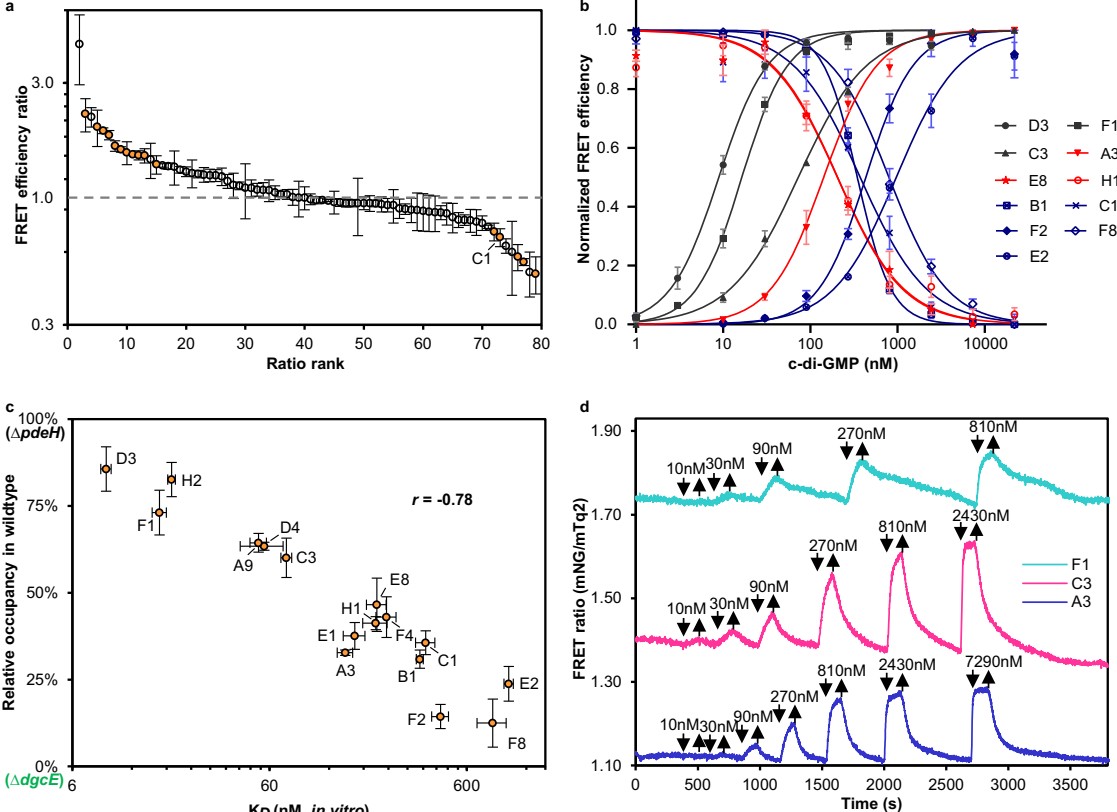

**Fig. 2 | Establishment and characterization of the biosensor toolbox. a** Ranked ratio of the FRET efficiency, measured using flow cytometry, for each biosensor in $\Delta pdeH$ vs $\Delta dgcE$ strains. Dashed line indicates no difference in FRET between the two strains. $n$ = 3 or 4 biological replicates. Data are presented as mean ± SD. Orange-labeled data points indicate biosensors chosen for the toolbox. **b** Dose responses of selected biosensors chosen for the toolbox, determined by flow cytometry upon incubation of permeabilized biosensor-expressing $\Delta dgcE$ cells with indicated concentrations of c-di-GMP. Data are presented as mean ± SEM. $n$ = 3, 4 or 6 biological replicates. The curves for the other five biosensors in the toolbox are shown in Supplementary Fig. 4a. **c** $K_D$ values determined from dose-response

measurements in permeabilized cells as in (b) and Supplementary Fig. 4a, plotted against the relative biosensor occupancy in the intact wildtype relative to $\Delta pdeH$ and in $\Delta dgcE$ cells (see Supplementary Fig. 6). $n$ = 3, 4 or 6 biological replicates for $K_D$ values and $n$ = 3 biological replicates for the occupancy values. Data are presented as mean ± SEM. Pearson correlation coefficient $r$ is shown. **d** Ratiometric FRET measurement of biosensor response to stepwise addition and subsequent removal of indicated concentrations of c-di-GMP, as indicated by arrows, in a flow chamber for selected biosensors in permeabilized $\Delta dgcE$ cells. Source data for (**a**–**d**) are provided as a Source Data file.

observation that even the high-affinity biosensors were not fully saturated in wildtype cells indicates that their c-di-GMP levels under our conditions are lower than previously reported[14].

We also measured cellular c-di-GMP dynamics of several biosensors using ratiometric microscopy in a flow chamber[37,38], where permeabilized $\Delta dgcE$ cells were exposed to step-like stimulation by different concentrations of c-di-GMP (Fig. 2d). In contrast to flow cytometry, where each point in the c-di-GMP-binding curve was measured at an equilibrium state, this microscopy measurement of FRET allows observation of both the dose-dependence and kinetics of the biosensor responses. All three tested biosensors showed responses within the concentration range expected based on their $K_D$ values. Of note, the amplitudes of responses to sub-saturating c-di-GMP concentrations in these measurements were likely moderately underestimated, because of the incomplete equilibration of the biosensors with the ligand during the limited time period prior to c-di-GMP removal. While all biosensors showed rapid c-di-GMP binding, the apparent dissociation rate varied between 0.11 min⁻¹ (F1) and 0.83 min⁻¹ (A3), demonstrating an expected negative correlation with the biosensor affinity, as already observed for previously published biosensors[10].

Finally, we demonstrated the applicability of our FRET biosensors for monitoring c-di-GMP in growing bacterial cells at single-cell resolution. For that, *E. coli* cells expressing FRET biosensor were grown in

microfluidic channels that were supplied with conditioned TB medium collected from an *E. coli* batch culture to mimic the early-stationary growth phase (see Methods). Since c-di-GMP levels in planktonic *E. coli* cells are low[14], we used a high-affinity F1 biosensor ($K_D$ ~ 17 nM). We observed high temporal dynamics and population heterogeneity of c-di-GMP levels in growing *E. coli* cells, which did not appear to be correlated with the cell cycle (Supplementary Fig. 7a). Notably, here we used the FRET ratio as a proxy for single-cell c-di-GMP levels, as done previously[34], since in these microscopy experiments the FRET ratio was not strongly affected by expression variability across individual cells (Supplementary Fig. 7b). In the wildtype population, c-di-GMP levels apparently varied across the entire responsive range of the F1 biosensor (~10–100 nM; Fig. 2b), yielding a range of FRET ratio values from low FRET comparable to the $\Delta dgcE$ strain to high FRET as observed in the $\Delta pdeH$ strain, where the c-di-GMP levels are expected to reach or exceed biosensor's saturation[14] (Supplementary Fig. 7c). For comparison, we also performed a microfluidics experiment with the low-affinity C1 biosensor, which is based on the same *Salmonella* YcgR as the previously published FRET biosensor[12,13]. The C1 biosensor was much less efficient in resolving c-di-GMP heterogeneity (Supplementary Fig. 7d), confirming that c-di-GMP levels in most *E. coli* cells under these conditions remained below the C1 sensitivity range (~100 nM; Supplementary Fig. 4b). Although the underlying causes of this c-di-GMP dynamics in *E. coli* remain to be investigated, these results

## Table 1 | Toolbox of FRET-based c-di-GMP biosensors

| | Biosensor | $K_D$ values in permeabilized cells (nM)[a] | $K_D$ values of purified protein (nM)[b] | $K_D$ values in the literature (nM) | FRET efficiency ratio in vivo[c] | FRET efficiency in $\Delta dgcE$ in vivo[d] |
|---|---|---|---|---|---|---|
| High affinity | D3 (*Syntrophothermus lipocalidus*) | 8.9 ± 1.3 | 20.6 ± 4.6 | | 2.0 ± 0.3 | 19.0% |
| | F1 (*Thermobrachium celere*) | 16.6 ± 2.7 | 37.5 ± 6.9 | | 1.9 ± 0.1 | 27.1% |
| | H2 (*Thermoanaerobacterales*) | 19.1 ± 1.7 | | <50[e] | 1.5 ± 0.2 | 18.4% |
| Medium affinity | A9 (*Massilia sp.*) | 52.7 ± 8.7 | | | 1.5 ± 0.1 | 34.6% |
| | D4 (*Pelotomaculum thermopropionicum*) | 56.4 ± 23.9 | | <50[e] | 1.4 ± 0.1 | 26.9% |
| | C3 (*Halothermothrix orenii*) | 73.0 ± 8.0 | 64.7 ± 18.8 | | 2.2 ± 0.4 | 19.2% |
| | A3 (*Thermincola potens*) | 145.1 ± 31.9 | 124.2 ± 7.3 | | 1.6 ± 0.1 | 25.0% |
| | E1 (*Symbiobacterium thermophilum*) | 162.1 ± 35.4 | | | 1.5 ± 0.0 | 22.9% |
| Low affinity | H1 (*Escherichia coli*) | 207.2 ± 58.6 | | 50-350[e] | 0.5 ± 0.1 | 31.1% |
| | E8 (*Methylotenera mobilis*) | 209.8 ± 41.6 | | | 0.5 ± 0.0 | 18.0% |
| | F4 (*Thermacetogenium phaeum*) | 235.0 ± 47.5 | | | 1.6 ± 0.0 | 18.3% |
| | B1 (*Pseudomonas putida* strain) | 346.6 ± 28.2 | | 104-165 [f] | 0.7 ± 0.1 | 34.5% |
| | C1 (*Salmonella typhimurium*) | 371.2 ± 72.1 | 384.1 ± 24.5 | 191-195 [g] | 0.7 ± 0.1 | 33.6% |
| | F2 (*Bacillus sporothermodurans*) | 441.0 ± 74.7 | | | 1.5 ± 0.1 | 30.2% |
| | F8 (*Pseudomonas putida*) | 812.4 ± 236.9 | | | 0.6 ± 0.1 | 35.9% |
| | E2 (*Natranaerobius thermophilus*) | 978.5 ± 92.1 | | | 1.8 ± 0.1 | 14.2% |

[a-c]Data are presented as mean ± SD. *n* = 3, 4 or 6 biological replicates; [c]Ratio between FRET efficiencies in $\Delta pdeH$ and $\Delta dgcE$ strains. Ratios higher than 1 or lower than 1 indicate opposite conformational changes upon c-di-GMP binding; [d]Mean values are presented. *n* = 3 or 4 biological replicates; [a-d]Source data are provided as a Source Data file; [e]Determined using BRET[25,36]. [f]Determined using isothermal titration calorimetry (ITC)[35]. [g]Determined using FRET[12,13].

confirm the suitability of our FRET biosensors for capturing cellular c-di-GMP dynamics and highlight the advantage of our toolbox that enables selection of a c-di-GMP biosensor with a suitable affinity.

### Mapping genetic network affecting c-di-GMP levels

The c-di-GMP regulatory networks in bacteria are highly complex and only partially understood even in model organisms including *E. coli*[39]. We thus sought to establish a methodology for a genome-wide mapping of c-di-GMP regulatory networks, utilizing our FRET biosensors that provide direct readout of cellular c-di-GMP levels in different concentration regimes, in combination with transposon mutagenesis and cell sorting (FRET-To-Sort). Transposon mutagenesis coupled to next-generation sequencing (TnSeq) is a powerful approach to annotate gene functions on a genome-wide scale in bacteria[40,41], but up to now it has not been used in combination with bacterial FRET biosensors.

We utilized a recently described *E. coli* Tn5 transposon library with random barcodes for transposon-site sequencing (RB-TnSeq), which enables easy quantification of the mutant abundance within bacterial population[26,40]. To select mutants with reduced c-di-GMP levels compared to the wildtype, we used the highest-affinity D3 biosensor, with $K_D$ value below the reported c-di-GMP concentration in planktonic *E. coli* cells[14]. This biosensor enables sensitive resolution of differences in the levels of c-di-GMP below that of the wildtype (Supplementary Fig. 4b and Supplementary Fig. 6a). For the sorting of mutants with elevated levels of c-di-GMP, we instead used the medium-affinity C3 biosensor, which enables better resolution in the range above the wildtype level of c-di-GMP.

The Tn5 libraries transformed with these biosensors were subjected to three sequential cycles of fluorescence activated cell sorting (FACS) based on FRET, to enrich for mutants with aberrantly high or low FRET, respectively (Fig. 3a). To validate our selection protocol, enriched mutants obtained from each cycle of sorting were individually grown and analyzed using flow cytometry. The results indicate a progressive shift in the distribution of FRET efficiencies towards higher or lower c-di-GMP levels following sorting (Fig. 3b,c). We next performed next-generation sequencing (NGS) on samples after two (×2)

and three (×3) sorting cycles and observed a largely reproducible enrichment of mutations with either high or low c-di-GMP levels (Supplementary Fig. 8a,b). Given that the selection for mutants with high c-di-GMP levels was sufficiently effective already after two sorting cycles (Fig. 3b), we primarily used the data from the second sorting for our subsequent analysis of high c-di-GMP mutants, to minimize potential unspecific mutant loss due to repeated growth selection. The selection for low c-di-GMP mutants was less effective (Fig. 3c), and we thus primarily used the data from the third sorting for the analysis of mutations showing low c-di-GMP levels. As a control, a small number of individual clones with high (comparable to or higher than in $\Delta pdeH$) or low (comparable to or lower than in $\Delta dgcE$) c-di-GMP levels (Fig. 3b,c) were individually genotyped by Sanger sequencing of barcodes[40]. This Sanger analysis validated a number of the NGS-selected mutations, and it also identified several additional mutations showing either high or low c-di-GMP levels (Supplementary Fig. 8c,d). These additional mutants identified by Sanger but not by NGS analysis are likely mutants with reduced growth, because such mutants may be underrepresented or absent in the NGS-enriched subpopulations that had to be passaged through an additional overnight growth step to obtain sufficient material for sequencing.

Overall, our analysis identified multiple mutations enriched for high or low c-di-GMP levels, which were largely consistent between both rounds of sorting (Tables 2–3, Supplementary Fig. 8, Supplementary Tables 2–3, and Supplementary Data 2 and 3). We observed enrichment of several functional networks of mutants with respectively elevated (Fig. 3d) or reduced (Fig. 3e) c-di-GMP levels. These networks were identified using the STRING database, which integrates all known and predicted protein–protein associations[42], and functional cluster analysis was performed using DAVID Bioinformatics[43]. The most prominent group of mutants with high c-di-GMP mapped to flagellar class II genes. Mutations in these genes are known to inhibit activity of the flagellar sigma factor FliA that drives transcription of *pdeH*[1,44] (see below). We further observed enrichment of *pdeH* itself, as well as of the mutation in *yfiR* encoding a putative periplasmic protein that was reported to inhibit the cyclase DgcN[45,46] (Fig. 3d). Both mutations are expected to upregulate the levels of c-di-GMP, thus

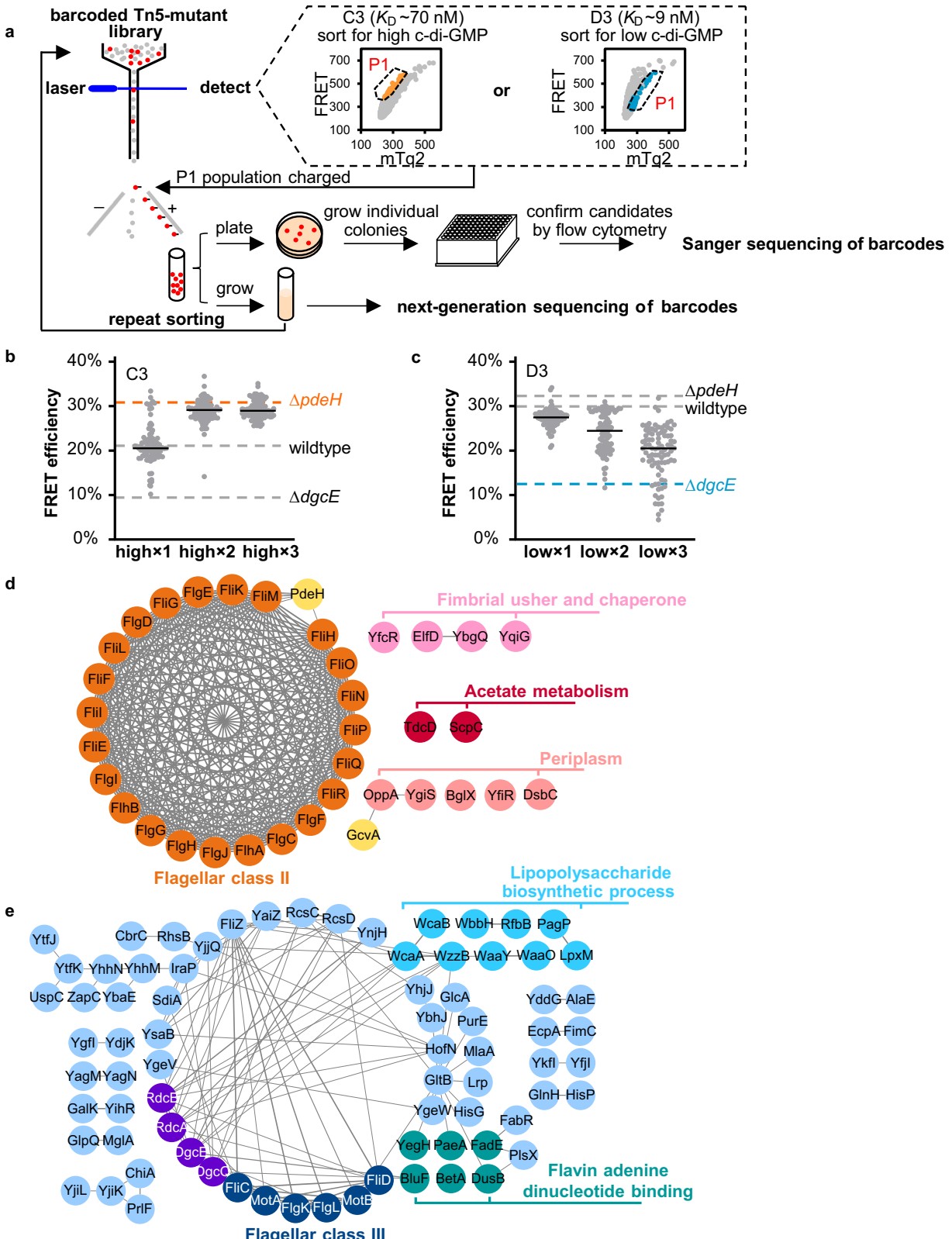

validating the effectiveness of our FRET-To-Sort selection protocol. Notably, no mutations in genes encoding other PDEs were enriched, suggesting that their contributions to c-di-GMP degradation under our conditions are minor. Another group of enriched mutations included genes encoding fimbrial chaperone ElfD and fimbrial usher proteins YbgQ, YfcR and YqiG (Fig. 3d). Mutations were also identified in genes encoding several periplasmic proteins, including disulfide isomerase

DsbC, periplasmic binding protein YgiS and OppA, and glycoside hydrolase BglX. In addition to periplasmic proteins, enriched mutations affect multiple cell envelope proteins, such as the outer membrane porins OmpW, transcriptional regulators affecting expression of cell wall (MraZ)[47], proteins generating enterobacterial common antigen (WecC and WecH), and the regulator of the cell-envelope stress response and outer membrane vesicle production HicB that is part of

**Fig. 3 | FRET-To-Sort application for genome-wide identification of c-di-GMP regulatory genes. a** Schematics of FRET-To-Sort, which is based on enrichment and subsequent identification of mutations from a randomly barcoded Tn5 transposon library through cell sorting based on the FRET signal. **b** Distributions of FRET values for 90 (high×1), 93 (high×2) and 89 (high×3) mutant colonies each, selected after the first (×1), second (×2) and third (×3) sorting for mutants with elevated c-di-GMP levels, measured using flow cytometry. The C3 biosensor ($K_D$ ~ 70 nM) was utilized for these experiments. Selected mutants confirmed to exhibit c-di-GMP levels comparable to those in the $\Delta pdeH$ strain (indicated by dashed line) or higher were genotyped by Sanger sequencing. **c** Distributions of FRET values for 93 (low×1), 92 (low×2) and 88 (low×3) mutant colonies each after the first (×1), second (×2) and third (×3) sorting for mutants with reduced c-di-GMP levels, measured

using flow cytometry. The D3 biosensor ($K_D$ - 9 nM) was utilized for these experiments. Selected mutants confirmed to exhibit c-di-GMP levels comparable to those in the $\Delta dgcE$ strain (indicated by dashed line) or lower were genotyped by Sanger sequencing. The networks and functional enrichment of mutants with elevated (**d**) or reduced (**e**) c-di-GMP levels identified by NGS and Sanger sequencing. The networks were generated using the STRING database version 11.5 and visualized using Cytoscape (version 3.9.1, NHGRI). The thicker edge width of the links means higher confidence of the proposed functional association between two proteins. The significant functional enrichment annotated using DAVID[43] ($P \leq 0.05$, modified Fisher's Exact test) is shown using colored text. Source data for b and c are provided as a Source Data file.

the toxin-antitoxin system[48] (Table 2, Supplementary Table 2). These indicate that defects in the cell envelope may be perceived as stress signals to elevate c-di-GMP levels. Finally, several mutations with high c-di-GMP were mapped to the acetate metabolism (Fig. 3d).

Among mutants with reduced levels of c-di-GMP, enriched were genes encoding DgcE and DgcQ, consistent with DgcE being the major cyclase in *E. coli*, and suggesting that DgcQ also significantly contributes to cellular c-di-GMP levels under our conditions. Also enriched were genes encoding the DgcE activator system that consists of RdcA and RdcB[49], as well as the gene encoding YnjH that exhibits strong association with this system (Fig. 3e). YjjQ, SdiA and FliZ are known to negatively regulate expression of the *flhDC* operon[50–52] that encodes the master regulator of flagellar synthesis, and the inactivation of their genes likely reduces c-di-GMP levels through the downstream activation of PdeH expression. We also observed enrichment of mutations in genes encoding proteins involved in lipopolysaccharide and lipid A biosynthesis and in flavin adenine dinucleotide (FAD) binding. Another prominent groups, although these genes did not show functional clustering, were genes associated with the general stress response. These include genes encoding the anti-adaptor protein IraP stabilizing RpoS[53], which is the master regulator of the general stress response and positively regulates c-di-GMP production in *E.coli*[1], the Rcs phosphorelay RcsCD that activates RpoS[54,55], the universal stress protein UspC, a stringent response modulator YtfK, proteins in oxidative stress response YhjJ and YbhJ, and the blue-light stress sensor BluF that is known to impact biofilm formation[56,57]. We also observed that mutations in genes encoding metabolic transcriptional regulators GltB, Lrp, HofN, HisG, GlcA, PunR and PurE and general chromosome organizing protein HU showed low c-di-GMP levels (Fig. 3e, Table 3, Supplementary Data 3 and Supplementary Table 3).

**Dual effect of mutations in flagellar genes on c-di-GMP levels**
Surprisingly, another group prominently enriched under selection for low c-di-GMP levels were mutations in flagellar class III genes, including *motA* and *motB* that encode the stator components of flagellar motor and *fliC* that encodes the major subunit of flagellar filament (Fig. 3e). This was opposite to the impact of mutations in flagellar class II genes (Fig. 3d), indicating a more complex interplay between motility and c-di-GMP regulation than previously recognized, which prompted us to further elucidate the underlying regulation.

As mentioned above, one mechanism connecting mutations in flagellar gene expression to c-di-GMP regulation is mediated through their impact on the expression of *pdeH*. Defects in the assembly of flagellar basal body and export apparatus (encoded by class II genes) preclude secretion of the anti-sigma factor FlgM[44], a negative regulator of FliA, and thus lead to inhibition of FliA activity and reduced PdeH expression. To confirm this mechanism of c-di-GMP regulation, we constructed deletions of two representative class II genes, encoding components of flagellar basal body (FliG) and export apparatus (FliH), in both the wildtype and $\Delta flgM$ backgrounds, and used the medium-affinity C3 biosensor to measure c-di-GMP levels in these strains. Consistent with our enrichment analysis using the same biosensor,

$\Delta fliG$ and $\Delta fliH$ cells showed largely elevated c-di-GMP levels compared to the wildtype (Fig. 4a). In contrast, no increase but rather a decrease in c-di-GMP level was observed when *fliG* and *fliH* were deleted in the $\Delta flgM$ background. Thus, positive effect of these mutations on c-di-GMP level indeed depends on the presence of FlgM.

We then investigated whether the effect of class III gene mutations might also be mediated by FlgM. However, similar reduction in c-di-GMP levels following *motA* and *fliC* deletions was observed in the wildtype and in the $\Delta flgM$ background, suggesting that the effects of these mutations are FlgM-independent (Fig. 4b). Here, we again used the high-affinity D3 biosensor to study strains with low c-di-GMP levels. Next, we tested whether the reduction of c-di-GMP levels by these class III gene mutations might require the activation of PdeH or the inactivation of DgcE. The effects of *motA* and *fliC* deletions could still be observed in $\Delta pdeH$ strain (Fig. 4c), ruling out a PdeH-dependent mechanism. Although the high-affinity D3 biosensor was apparently saturated in the absence of *pdeH* (Supplementary Fig. 9), these differences could be well resolved using the medium-affinity C3 biosensor.

To assess the involvement of DgcE, we instead investigated the impact of *motA* and *fliC* mutations in $\Delta pdeH \Delta dgcE$ and $\Delta pdeH dgcE^{GGAAF}$ strains. This was done because in the presence of the dominant PDE PdeH, the levels of c-di-GMP are already very low in $\Delta dgcE$[14], or in $dgcE^{GGAAF}$ strain carrying mutation in the catalytically active site[49], and we reasoned that it would be difficult to detect any further reduction of c-di-GMP, even using the high-affinity D3 biosensor. Since $\Delta pdeH \Delta dgcE$ and $\Delta pdeH dgcE^{GGAAF}$ strains carry inactivatory mutations in both, the major DGC and the major PDE of *E. coli*, they are known to have intermediate c-di-GMP levels that are maintained by other DGCs and PDEs[14]. The deletion of *motA* or *fliC* did not result in a significant alteration of c-di-GMP levels in these background strains (Fig. 4c). Given that the involvement of PdeH was already ruled out (see above), these results suggest that mutations in flagellar class III genes affect c-di-GMP production via DgcE. Confirming the physiological impact of the observed c-di-GMP regulation, these results were consistent with the c-di-GMP-dependent expression of curli genes, which encode a key proteinaceous extracellular biofilm matrix component of *E. coli*[2,28] (Supplementary Fig. 10).

We hypothesized that one possible physiological impact of the class III flagellar mutations in *E. coli* could be the perturbation of the proton motive force (PMF). As flagellar rotation is driven by dissipation of PMF through the MotA/MotB stator complexes engaged with the flagellar rotor[58], deletion of *motA* or *motB* will abolish this proton flux and thus result in the elevation of cellular PMF. Lack of the flagellar filament, due to mutations in either *fliC* or in genes encoding other class III flagellar components that are required for proper filament assembly, can elevate PMF as well. One explanation is that the filament-less motors experience much lower mechanical load, which is known to result in the disengagement of stators from the motor and a consequent reduction of proton flux[59]. Another reason is that the filament assembly is PMF-dependent[60] and therefore the absence of FliC might decrease the PMF dissipation. Indeed, *motA* and *fliC* mutants exhibited

**Table 2 | Mutations with elevated c-di-GMP levels identified by NGS analysis**

| Insertion location[b] | Mutations enriched for elevated c-di-GMP levels[a] | | | |
|---|---|---|---|---|
| | Gene product description[b] | Log2 (fold change)[c] | | Confirmed by Sanger sequencing[d] |
| | | ×2 | ×3 | |
| gcvA | DNA-binding transcriptional dual regulator | 11.8 | NA[e] | |
| fliO | flagellar biosynthesis protein | 9.2 | 11.9 | |
| fliI_**fliJ** | fliI: flagellar export ATPase; fliJ: flagellar biosynthesis protein | 9.1 | 12.1 | Yes |
| pdeH | cyclic di-GMP phosphodiesterase | 8.9 | NA | |
| moeA promoter | molybdopterin molybdotransferase | 8.8 | 11.6 | Yes |
| dsbC | protein disulfide isomerase | 8.3 | 11.7 | Yes |
| wecC | UDP-N-acetyl-D-mannosamine dehydrogenase | 7.9 | NA | |
| **chbF**_chbR | chbF: monoacetylchitobiose-6-phosphate hydrolase<br>chbR: DNA binding transcriptional dual regulator | 7.8 | 11.3 | Yes |
| mdaB promoter | NADPH:quinone oxidoreductase | 7.6 | 11.1 | Yes |
| yfcR | putative fimbrial protein | 7.4 | NA | |
| pepQ | Xaa-Pro dipeptidase | 7.2 | 10.9 | Yes |
| ydcO | putative transport protein | 6.8 | 11.3 | Yes |
| flgF | flagellar basal-body rod protein | 6.6 | 9.5 | Yes |
| galM | galactose-1-epimerase | 5.8 | 8.9 | |
| ymfE | e14 prophage; uncharacterized protein | 5.6 | 10.5 | |
| mraZ promoter | DNA-binding transcriptional repressor | 5.3 | 9.4 | |
| wecH | O-acetyltransferase | 5.0 | 6.8 | Yes |
| elfD | putative fimbrial chaperone | 4.8 | 8.7 | |
| fliE promoter | flagellar protein | 4.7 | 5.1 | Yes |
| glpC | anaerobic glycerol-3-phosphate dehydrogenase subunit C | 4.6 | NA | |
| flhB | flagellar biosynthesis protein | 4.6 | 4.2 | Yes |
| ycgH | putative transporter component, N-terminal fragment | 4.6 | 8.9 | Yes |
| yfjP | CP4-57 prophage; putative GTP-binding protein | 4.6 | 9.1 | Yes |
| flgG | flagellar basal-body rod protein | 4.5 | 4.0 | Yes |
| insI1 | IS30 transposase | 4.5 | 8.6 | |
| proB promoter | glutamate 5-kinase | 4.5 | 9.3 | |
| fliL | flagellar protein | 4.5 | 3.1 | Yes |
| flgE | flagellar hook protein | 4.4 | 4.1 | Yes |
| fliN | flagellar motor switch protein | 4.3 | 4.0 | Yes |
| gntT | high-affinity gluconate transporter | 4.3 | 7.9 | Yes |
| ygjV | inner membrane protein | 4.3 | 3.9 | Yes |
| uidA | β-D-glucuronidase | 4.2 | 7.9 | |
| fliH | flagellar biosynthesis protein | 4.1 | 4.0 | Yes |
| flgI | flagellar P-ring protein | 4.1 | 4.0 | Yes |
| flgJ | putative peptidoglycan hydrolase | 4.0 | 3.9 | Yes |
| cyuA | putative L-cysteine desulfidase | 3.9 | 8.7 | Yes |
| yabQ_rluA | yabQ: protein YabQ<br>rluA: 23S rRNA pseudouridine$^{746}$ and tRNA pseudouridine$^{32}$ synthase | 3.9 | 3.2 | Yes |
| fliR | flagellar biosynthesis protein | 3.9 | 4.0 | Yes |
| scpC | propionyl-CoA:succinate CoA transferase | 3.9 | 4.3 | Yes |
| **osmF**_bglX | osmF: glycine betaine ABC transporter periplasmic binding protein<br>bglX: putative periplasmic glycoside hydrolase | 3.8 | 3.4 | Yes |
| ompW | outer membrane protein W | 3.8 | 8.4 | Yes |
| fliQ | flagellar biosynthesis protein | 3.8 | 4.4 | Yes |
| yqiG | putative outer membrane usher protein | 3.6 | 7.8 | Yes |
| yjjV | putative metal-dependent hydrolase | 3.6 | 7.6 | Yes |

[a]Mutants enriched in the second (×2) sorting cycle, with log2 (fold change) > 3.5 and $P \le 0.05$, are listed. Mutants with $2 \le \text{log2 (fold change)} \le 3.5$ enrichment in the second (×2) sorting cycle are displayed in Supplementary Data 2. Enrichment of the same mutants in the third (×3) sorting cycle is shown for a reference. $P$ values were calculated using unpaired $t$-test with Bayesian regularization (Bayesian prior set to 6) implemented in the Cyber-T web server[91]. Source data are provided as a Source Data file. Exact $P$ values are reported in the Source Data file.

[b]Underline indicates that the insertion is located between two neighbor genes. Genes labeled in bold indicate that the insertions are located upstream of them. All gene names are italicized by convention.

[c]The fold change is calculated as the ratio of enrichment in high sorting sample over enrichment in low sorting sample.

[d]"Yes" indicates that mutants enriched in the NGS analysis were also genotyped by Sanger sequencing. Otherwise, blank.

[e]Not applicable because the gene has a low number of reads (≤ 100).

**Table 3 | Mutations with reduced c-di-GMP levels identified by NGS analysis**

| Insertion location[b] | Mutations enriched for reduced c-di-GMP levels[a] | Log2 (fold change)[c] | | Confirmed by Sanger sequencing[d] |
|---|---|---|---|---|
| | Gene product description[b] | ×2 | ×3 | |
| argD_yhfK | argD: N-acetylornithine aminotransferase / N-succinyldiaminopimelate aminotransferase; yhfK: putative transporter | −11.4 | −12.1 | |
| rlmJ | 23S rRNA m$^6$A2030 methyltransferase | −8.8 | −11.2 | |
| yqgC | protein YqgC | −8.4 | −11.2 | |
| fadE promoter | acyl-CoA dehydrogenase | −10.7 | −11.0 | |
| chiA promoter | endochitinase | −9.0 | −10.7 | |
| ybfL | putative transposase | −7.0 | −10.3 | |
| ecpA promoter | common pilus major subunit | −12.7 | −10.3 | |
| ynbA | CDP-alcohol phosphatidyltransferase domain-containing protein | −6.3 | −10.0 | |
| yegS_gatD | yegS: lipid kinase gatD: galactitol-1-phosphate 5-dehydrogenase | −11.1 | −9.8 | |
| efp_**ecnA** | efp: protein chain elongation factor; **ecnA**: entericidin A lipoprotein | −7.3 | −9.8 | |
| ytfJ | PF09695 family protein | −5.8 | −9.8 | |
| yhjJ | peptidase M16 family protein | −8.0 | −9.7 | Yes |
| rluC | 23S rRNA pseudouridine$^{955/2504/2580}$ synthase | −6.7 | −9.6 | |
| essD promoter | DLP12 prophage; putative phage lysis protein | −6.8 | −9.5 | |
| kbaY | tagatose-1,6-bisphosphate aldolase 1 | −6.2 | −9.4 | |
| lpxM | lipid A biosynthesis myristoyltransferase | −5.9 | −9.4 | |
| hupA | protein HUαα | −8.3 | −9.4 | |
| flgL | flagellar hook-filament junction protein 2 | −4.6 | −9.3 | |
| sgbH | 3-keto-L-gulonate-6-phosphate decarboxylase | −7.1 | −9.3 | |
| ompG_ycjW | ompG: outer membrane porin G; ycjW: DNA-binding transcriptional repressor | −5.9 | −9.2 | |
| yegV | putative σ$^{54}$-dependent transcriptional regulator | −5.2 | −9.2 | |
| flgK | flagellar hook-filament junction protein 1 | −6.9 | −9.1 | Yes |
| idnT | L-idonate/5-ketogluconate/gluconate transporter | −5.1 | −9.1 | |
| paeA | putative cadaverine/putrescine exporter | −6.5 | −9.0 | |
| prlF | antitoxin | −5.9 | −9.0 | |
| motB | motility protein B | −7.1 | −9.0 | Yes |
| betA | choline dehydrogenase | −5.2 | −8.9 | |
| **yfaY**_yfaZ | **yfaY**: IPR008135 CinA family protein; yfaZ: putative porin | −9.2 | −8.9 | |
| fabB promoter | 3-oxoacyl-[acyl carrier protein] synthase 1 | −5.0 | −8.9 | |
| **araE**_kduD | **araE**: arabinose:H$^+$ symporter kduD: putative 2-keto-3-deoxy-D-gluconate dehydrogenase | −4.9 | −8.8 | |
| yfdF | protein YfdF | −6.1 | −8.8 | |
| yhhM promoter | DUF2500 domain-containing protein | −9.0 | −8.8 | |
| yhhN promoter | PF07947 family protein | −9.0 | −8.8 | |
| yihN | putative transporter | −5.0 | −8.7 | |
| glnH promoter | L-glutamine ABC transporter periplasmic binding protein | −7.8 | −8.7 | |
| punR | DNA-binding transcriptional activator | −8.3 | −8.6 | |
| nimT | 2-nitroimidazole exporter | −8.1 | −8.6 | |
| zapC | cell division protein | −9.2 | −8.6 | |
| plsX | putative phosphate acyltransferase | −8.8 | −8.6 | |
| yagM | CP4-6 prophage | −4.6 | −8.5 | |
| ypfJ | uncharacterized protein | −4.0 | −8.5 | |
| yfhR | putative peptidase | −5.8 | −8.5 | |
| melR promoter | DNA-binding transcriptional dual regulator | −8.3 | −8.5 | |

[a]Mutants enriched in the third (×3) sorting cycle, with log2 (fold change) ≤ -8.5 and $P$ ≤ 0.05, are listed. Mutants with -8.5 < log2 (fold change) ≤ -3 enrichment in the third (×3) sorting cycle are displayed in Supplementary Data 3. Enrichment of the same mutants in the second (×2) sorting cycle is shown for a reference. $P$ values were calculated using unpaired $t$-test with Bayesian regularization (Bayesian prior set to 6) implemented in the Cyber-T web server[91]. Source data are provided as a Source Data file. Exact $P$ values are reported in the Source Data file.
[b]Underline indicates that the insertion is located between two neighbor genes. Genes labeled in bold indicate that the insertions are located upstream of them. All gene names are italicized by convention.
[c]The fold change is calculated as the ratio of enrichment in high sorting sample over enrichment in low sorting sample.
[d]"Yes" indicates that mutants enriched in the NGS analysis were also genotyped by Sanger sequencing. Otherwise, blank.

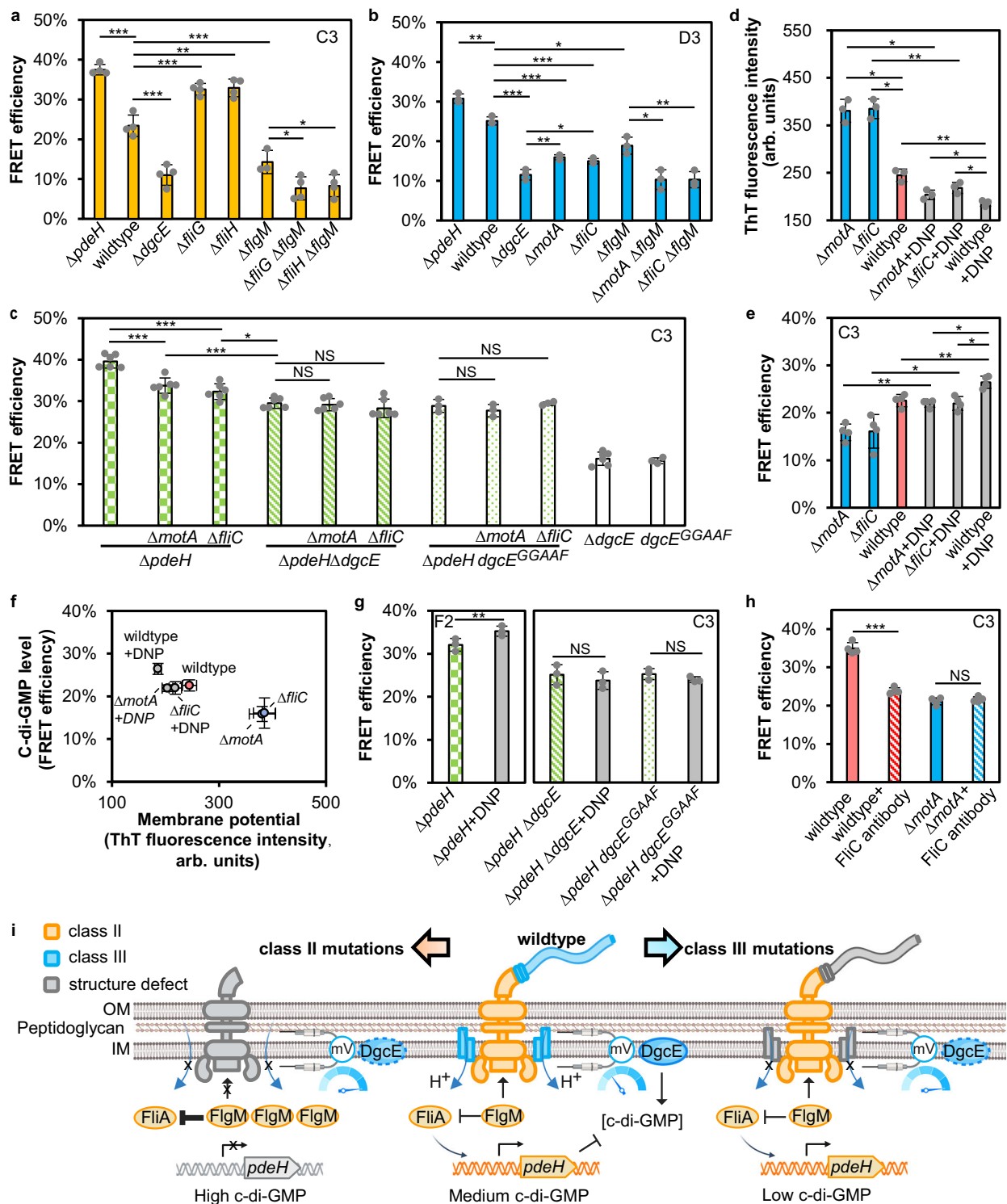

higher membrane potential (the major component of PMF[58]) compared to the wildtype, measured using fluorescence of the potential-sensitive dye Thioflavin T (ThT)[61] (Fig. 4d). The control treatment with the protonophore dinitrophenol (DNP)[62] decreased membrane potential in all strains, as expected. Mirroring these changes in the membrane potential, treatment with DNP resulted in elevated c-di-GMP levels in all strains (Fig. 4e), which confirmed the inverse relationship between the membrane potential and c-di-GMP levels (Fig. 4f). This impact of changes in the membrane potential on c-di-GMP levels depended on DgcE, as the upregulation of c-di-GMP by DNP was observed in the wildtype (Fig. 4e) and the ΔpdeH stain but not in

the ΔpdeH ΔdgcE strain (Fig. 4g). Dependent on the background-specific basal c-di-GMP levels, we either used the low-affinity F2 biosensor or the medium-affinity C3 biosensor. Notably, DNP only partially dissipated the membrane potential in ΔmotA and ΔfliC strains (Fig. 4d), and therefore their c-di-GMP levels remained lower than those of the wildtype cells (Fig. 4e). The increased membrane potential in the absence of a functional flagellar motor largely, but apparently not entirely inhibited DgcE activity, since c-di-GMP levels in ΔmotA or ΔfliC strains remained slightly higher compared to ΔdgcE (Fig. 4b), and those in ΔpdeH ΔmotA or ΔpdeH ΔfliC backgrounds were slightly higher than those in ΔpdeH ΔdgcE (Fig. 4c).

**Fig. 4 | Opposite effects on c-di-GMP levels observed for mutations in flagellar class II and class III genes.** FRET biosensors used in respective experiments are indicated. Higher FRET efficiency indicates higher c-di-GMP levels. Impact of deletions of flagellar class II (**a**) and class III (**b**, **c**) genes on c-di-GMP levels. *P* values were calculated using unpaired two-tailed *t-test*, *n* = 3 or 4 biological replicates in (**a**), *n* = 3 biological replicates in (**b**) and *n* = 3 or 6 biological replicates in (**c**). **d** Impact of deletions of class III flagellar genes on the membrane potential, measured using the potential-sensitive ThT dye. Higher fluorescence intensity in arbitrary (arb.) units indicates higher membrane potential. Control DNP treatment reduces membrane potential. *P* values were calculated using paired two-tailed *t-test*, *n* = 3 biological replicates. **e** Impact of DNP and class III flagellar gene mutations on c-di-GMP levels. Note difference between biosensors used here and in (**b**). *P* values were calculated using paired two-tailed *t-test*, *n* = 4 biological replicates. **f** Relation between membrane potential and c-di-GMP levels, shown for indicated

strains/conditions. *n* = 3 biological replicates for membrane potential and *n* = 4 biological replicates for c-di-GMP level. **g** DgcE-dependent impact of DNP on c-di-GMP levels. *P* values were calculated using paired two-tailed *t-test*, *n* = 3 biological replicates. **h** Impact of antibody-inhibited flagellar rotation on c-di-GMP levels. Note that after antibody treatment cells had to be washed by centrifugation prior to flow cytometry measurements, which led to generally elevated basal FRET levels in all strains. *P* values were calculated using paired two-tailed *t-test*, *n* = 4 biological replicates. \*\*\**P* < 0.001; \*\**P* < 0.01; \**P* < 0.05; NS, no significant difference. Data are presented as mean ± SD. **i** Illustration of impact of mutations in flagellar class II and class III genes on c-di-GMP levels and the proposed underlying mechanisms. Created in BioRender. Wang, L. (2026) https://BioRender.com/66mu5d8. Exact *P* values for all statistical comparisons are reported in the Source Data file. Source data for (**a**–**h**) are provided as a Source Data file.

The increase of the membrane potential was also expectedly observed in Δ*fliG* and Δ*fliH* mutants, which lack flagella and motors and cannot dissipate PMF (Supplementary Fig. 11). This explains the negative impact of these class II gene mutations on c-di-GMP levels in the Δ*flgM* background (Fig. 4a; Δ*fliG* Δ*flgM* and Δ*fliH* Δ*flgM* compared to Δ*flgM*). In the absence of *flgM*, these mutations no longer affect the levels of c-di-GMP through PdeH expression but only through the membrane potential, similar to class III genes.

We hypothesized that such regulation of c-di-GMP by the membrane potential may provide a mechanosensory mechanism that enables *E. coli* to respond to immobilization of flagella. In order to test this, we incubated biosensor-expressing *E. coli* cells with the FliC-specific antibody that binds to flagellar filaments and inhibits their rotation. Indeed, this treatment not only led to inhibition of motility (Supplementary Fig. 12 and Supplementary Movies 1-2) but also caused a significant drop in c-di-GMP levels in wildtype cells (Fig. 4h). In contrast, antibody had no impact on c-di-GMP levels in Δ*motA* cells where flagella are already immobile. These results confirmed that the observed potential-dependent c-di-GMP regulation can provide a readout of flagellar rotation.

Such antibody-mediated jamming of flagellar rotation mimics mechanical cues associated with surface contact by swimming cells[63] or with swimming in high-viscosity media. Indeed, we observed that high medium viscosity reduced both, swimming velocity and c-di-GMP levels (Supplementary Fig. 13), consistent with the positive correlation between PMF dissipation by rotating motors and c-di-GMP levels. We next used confocal microscopy to monitor c-di-GMP levels (again, using the FRET ratio as a proxy) at the early stage of surface colonization and aggregate formation by *E. coli* cells. We observed that surface-attached cells had lower c-di-GMP levels than cells in aggregates floating above the surface (Supplementary Fig. 14 and Supplementary Movie 3). Interestingly, these findings in *E. coli* contradict to the prevailing paradigm of mechanosensing, which states that the inhibition of flagellar rotation should upregulate c-di-GMP levels[63–67].

## Discussion

Signaling mediated by c-di-GMP is a key determinant of transition between planktonic and sessile lifestyles in bacteria, but its complex regulation and dynamics remain only partly understood. Measuring c-di-GMP in living cells across different organisms and conditions is thus critical. However, up to now there is only a limited choice of real-time c-di-GMP biosensors[10–13], and all of them exhibit relatively low affinities and use different designs or fluorophores. Given that levels of c-di-GMP can vary widely between organisms and conditions[1,2,18,19,23] and can be very low[14–16], here we addressed these limitations by developing a toolbox of sixteen c-di-GMP FRET biosensors with the same design and variable affinities, including high-affinity biosensors. Since these biosensors function autonomously, they should be broadly applicable in various bacterial species, requiring only the levels of c-di-GMP in the

target specie to fall within the broad working range of our toolbox and the availability of an expression system.

To efficiently screen this biosensor toolbox and to apply it for investigation of c-di-GMP levels in different *E. coli* strains, we established a high-throughput flow-cytometry based FRET measurement assay, correcting the bleed-through from the donor emission and the direct excitation of the acceptor to accurately calculate FRET efficiency. We also demonstrated the application of established biosensors for time-resolved monitoring of single *E. coli* cells in a microfluidic device, uncovering large dynamic fluctuations in c-di-GMP levels. Thus, our toolbox should be readily applicable for standardized monitoring of c-di-GMP dynamics across population or single cells of different organisms and under different growth conditions, using either flow cytometry or fluorescence microscopy.

To further expand the applications of FRET biosensors in bacteria, we developed a FRET-To-Sort approach for genome-wide mapping of c-di-GMP regulatory networks. This approach combines the flow-cytometry based FRET measurement with cell sorting, and with RB-TnSeq that is a powerful approach to determine genetic interactions in bacteria[40,41]. Notably, while FRET biosensors in combination with FACS have been successfully used for genetic screening in mammalian cells[68], the potential of FRET-based cell sorting for such screens has not been previously explored in bacteria. Leveraging the advantages of our toolbox, we utilized two biosensors with different affinities to respectively identify mutations resulting in high or low c-di-GMP levels from the recently constructed *E. coli* barcoded transposon mutant library[26].

FRET-To-Sort effectively enriched genes known to regulate c-di-GMP in *E. coli*, including those encoding the major cyclase DgcE and its positive regulators RdcA and RdcB, and the major phosphodiesterase PdeH. The gene encoding the cyclase DgcQ was also enriched, consistent with its important contribution to the activation of biofilm matrix genes in *E. coli*[28]. Apart from the inhibitor of cyclase DgcN, no enrichment was observed for genes encoding other cyclases or phosphodiesterases in *E. coli*, indicating that these other enzymes make only minor contributions to the global pool of c-di-GMP under our conditions. We further observed enrichment of multiple genes known to affect the general stress response (e.g., *iraP* and *idlP* encoding proteins stabilizing RpoS[53,69]), stringent response (e.g., *ytfK*), and biofilm formation (e.g., *rcsC*, *rcsD* and *bluF*)[56,57,70], consistent with the expected relation of these processes to c-di-GMP regulation in *E. coli*[53,71]. In addition, we identified a number of genes with strong but previously little- or uncharacterized impact on c-di-GMP signaling. Among mutations causing elevated c-di-GMP levels were those in genes encoding components of Elf, Ybg, Yqi and Yfc fimbriae, whereas inactivation of Ecp and type-I fimbria lowered c-di-GMP levels. Besides fimbrial genes, a group of mutations affecting periplasm or lipopolysaccharide and lipid A metabolism also led to high or low c-di-GMP levels, suggesting that perturbation of the cell envelope and membrane composition might generally affect c-di-GMP signaling. Cell-

envelope stress is known to affect biofilm formation in *E. coli*[28,72] and in other bacteria[73]. How it leads to the c-di-GMP upregulation in *E. coli* remains to be investigated, although the observed sensitivity of c-di-GMP biosynthesis to the membrane potential could provide one possible mechanism. Nevertheless, cell-envelope stress can apparently lead to either increase or decrease in intracellular c-di-GMP levels, dependent on its type. In contrast, disruption of multiple genes encoding transcription factors that regulate nutrient utilization, as well as of several FAD binding proteins, consistently lowered c-di-GMP levels. The latter may be associated with the importance of the respiration process during biofilm formation[74].

The most prominent groups of genes affecting the c-di-GMP levels in *E. coli* were flagellar genes. Unexpectedly, we observed opposite effects for mutations in class II and in class III flagellar genes, revealing a dualistic function of flagella in the c-di-GMP regulation (Fig. 4i). The first level of control, consistent with the established antagonism between flagellar gene expression and c-di-GMP levels in *E. coli*[1,44], is primarily mediated by the FliA-dependence of *pdeH* expression. As a consequence of this transcriptional regulation, *pdeH* is repressed along with the other class III flagellar genes when FliA is not active, which happens when assembly of the flagellar basal body or export apparatus is prevented, and the anti-FliA factor FlgM is retained in the cytoplasm. This *pdeH* repression leads to reduced c-di-GMP degradation, thereby increasing its intracellular levels.

In addition to this expected antagonism between activity of the flagellar regulon and c-di-GMP levels[64], we also uncovered an opposite regulation, with inactivation of genes encoding flagellin FliC, the stator components MotA and MotB and other class III motility genes leading to a strong reduction in c-di-GMP levels. This is generally consistent with previous observations that functionality of the flagellar motor can affect the c-di-GMP production in different bacteria including *E. coli*[19,64,66,75]. We found that this effect was not dependent on FlgM or PdeH, but it required the presence of DgcE. Our analysis suggests that the downregulation of c-di-GMP by the class III flagellar gene mutations is due to the inhibition of the DgcE activity by elevated membrane potential (or PMF), because they reduce or abolish dissipation of the PMF though flagellar motor. Complementary, the DgcE activity is upregulated by reduced membrane potential (e.g., DNP treatment) demonstrating an overall DgcE-dependent inverse relationship between membrane potential (or PMF) and c-di-GMP levels.

Although the exact mechanism of PMF sensing by DgcE remains to be elucidated, our results suggest that regulation of c-di-GMP levels dependent on flagellar rotation may be utilized by *E. coli* cells for mechanosensing of surface contact or of other conditions that hinder flagellar rotation[76]. This is supported by our observation that the inhibition of flagellar rotation, either by antibody-mediated jamming, by high viscosity of the medium or by cell association with the surface, elevates membrane potential and results in lowering of c-di-GMP levels in *E. coli* cells. This also agrees with the previous observation that, during formation of *E. coli* static submerged biofilms, the c-di-GMP-dependent production of curli biofilm matrix primarily occurs within aggregates above the surface rather than in directly attached cells[9]. This contrasts with the common assumption that inhibited flagella rotation, for example, upon surface contact, should lead to increased c-di-GMP levels or biofilm formation[65–67]. These findings underscore the diversity of regulatory interplays between motility and c-di-GMP regulation[64].

Notably, the interplay between the load on flagellar motor and c-di-GMP levels may be complicated by the load-dependent recruitment of stator units[59]: While stators are apparently disengaged and conduct little proton flux in the absence of load, there is also low or no flux at high load when the motor rotation is hindered despite stator engagement. Consequently, both low and high motor load may lead to low levels of c-di-GMP, whereas maximal flux, and therefore highest

PMF dissipation and high c-di-GMP levels, are expected at intermediate loads, such as those experienced by free-swimming cells in low-viscosity media.

More generally, the combination of different-affinity biosensors from our toolbox and FRET-To-Sort established here should be readily extendable to investigate regulation of cellular levels of c-di-GMP across a variety of bacterial species where corresponding mutant libraries and expression systems are available, including important pathogens[77,78]. Moreover, FRET-To-Sort should be similarly applicable to any other existing FRET-based bacterial second messenger or metabolite biosensors[34].

# Methods

## Bacterial strains and plasmids
All strains and plasmid vectors used in this study are listed in Supplementary Table 4. *E. coli* strains were derived from the RpoS+ variant of W3110[79], by PCR-product based inactivation of chromosomal genes[80] using plasmid pSIJ8[81] to introduce gene deletions. The point mutation of *dgcE* (*dgcE*[GGAAF]) was generated using a four-primer/two-step PCR protocol[49]. Library of genes encoding YcgR-homologous PilZ-domain proteins[25] fused to mNeonGreen and mTurquoise2 was constructed and cloned into the pTrc99A vector using Gibson assembly[82]. Following previous study[25], we used linkers AACGGCAGCCCATGG between mNeonGreen and YcgR, and GAGCTCTACAGGCTG between YcgR and mTurquoise2. The linear backbone was produced using restriction enzymes NcoI and HindIII, and insert fragments were amplified by PCR. *E. coli ycgR* sequence was amplified from the *E. coli* W3110 genomic DNA, and the other 89 *ycgR* sequences were amplified from the published pET21/24-YNL-YcgR-91-mCherry plasmid library[25]. For the construction of plasmids expressing mNeonGreen alone and mTurquoise2 alone, respective genes were separately cloned into the pTrc99A vector or into pBAD33 vector. All oligonucleotide primers used for mutagenesis and plasmid construction are listed in Supplementary Data 3.

## Media and growth conditions
Cells were grown overnight in tryptone broth (TB; 1% tryptone, 0.5% NaCl) medium at 37 °C with shaking at 200 rpm to $OD_{600}$ ~ 2.0. Fifty microliters of overnight culture were diluted into 1 mL (for deep 96-well plates) or into 5 mL (for deep 24-well plates) fresh medium containing 20 μM isopropyl β-D-thiogalactoside (IPTG), for library screening or the $K_D$ measurements, respectively. For control cultures carrying plasmids separately co-expressing mTurquoise2 and mNeonGreen from pBAD33 and pTrc99A vectors, respectively, the medium further contained 0.01% arabinose. Cells were harvested after 4.5–5 h of incubation in a rotary shaker at 37 °C with shaking at 200 rotations per minute (rpm) to $OD_{600}$ ~ 0.6. Bacterial culture was then diluted or resuspended in an appropriate buffer prior to measurements FRET efficiency using flow cytometry or photobleaching microscopy. Where necessary, the culture medium was supplemented with antibiotics (100 μg/mL ampicillin, 34 μg/mL chloramphenicol, or 50 μg/mL kanamycin).

## Measurements of FRET efficiency using flow cytometry
The median fluorescence intensities of each channel were measured using BD LSRFortessa SORP cell analyzer (BD Biosciences, Germany) with BD FACSDiva software (version v8.0.1, BD). For each sample, a total of 30,000 events was acquired. As illustrated in Supplementary Fig. 1b−d, fluorescence intensities were measured in the donor channel ($I_1$; 445 nm excitation, 470/15 nm emission), FRET channel ($I_2$; 445 nm excitation, 520/35 nm emission) and acceptor channel ($I_3$; 488 nm excitation, 520/30 nm emission). The power of both, 445 nm and 488 nm lasers was set at 75 mW. The median background level of fluorescence intensity, obtained by measuring autofluorescence of the Δ*dgcE* strain carrying the empty pTrc99A vector, was always less than

2% of the signal in each channel and it was subtracted from each of the median fluorescence intensities.

The FRET efficiency was calculated from flow cytometric data by adapting a previously established approach[30], as follows (Supplementary Fig. 1b):

$$E = \frac{I_2 - S_2 I_3 - S_1 I_1}{I_2 - S_2 I_3 + (\alpha - S_1) I_1} \times 100\% \tag{1}$$

where $S_1$ corrects for the spectral bleed-through arising from the donor emission into the FRET channel, and $S_1$ equals to $I_2/I_1$, measured using a "donor-only" samples, i.e., cells expressing mTurquoise2 alone. $S_2$ corrects for the bleed-though in the FRET channel arising from the not negligible direct excitation of the acceptor at 445 nm, and $S_2$ equals to $I_2/I_3$, measured using "acceptor-only" samples, i.e., cells expressing mNeonGreen alone. The value of $\alpha$ is the calibration factor[83] relating the fluorescence intensity of an acceptor molecule in the $I_2$ channel to the intensity of a donor molecule in the $I_1$ channel for the mTurquoise2-mNeonGreen FRET pair for a given experimental setup. This value was determined as $\alpha = 0.739 \pm 0.081$ by calculating the ratio of the slope and the intercept from the plot of $y = \frac{1}{R_F - 1}$ vs $x = \frac{1}{R_1}$, where $R_1 = \frac{I_2 - S_2 I_3 - S_1 I_1}{I_1}$ and $R_F = \frac{I_2 - S_1 I_1}{S_2 I_3}$. Notably, given the number of available biosensor constructs, we were able to obtain the $\alpha$ calibration factor from the interpolation of 48 values from independent one-to-one donor/acceptor samples having different FRET efficiencies (Supplementary Fig. 15).

For measurements of $E$ values in intact cells, cultures were grown as described above. Fifty microliters of day culture were diluted into 1 mL tethering buffer (20 mM potassium phosphate, 0.1 mM EDTA, 1 μM methionine, 10 mM sodium lactate, pH 7), briefly vortexed to fully disrupt occasional cell aggregates, equilibrated for 1 h and vortexed again prior to measurements. The absence of aggregates, even in the $\Delta pdeH$ strain that has high levels of c-di-GMP, was confirmed using forward scatter (FSC) and side scatter (SSC) analysis (Supplementary Fig. 16). Flow cytometry data were exported using FlowJo (version 10.10.0, BD).

For measurements of $K_D$, $\Delta dgcE$ cells carrying biosensors were grown as described above, washed and resuspended in the assay buffer (50 mM potassium phosphate [pH 7.4], 100 mM KCl, 0.5 mM MgCl$_2$, 13 mM NaCl), in which ionic composition is similar to that of bacterial cytoplasm[84,85]. After adding 1% toluene, cell suspension was incubated at 24 °C with shaking at 300 rpm for 10 min, and diluted 1:20 into the assay buffer containing different concentrations of c-di-GMP. The mixture was vortexed and left at room temperature for at least 1 h to equilibrate before the measurement. The $K_D$ values were calculated from the c-di-GMP dose-response curves using allosteric sigmoidal model in GraphPad Prism (version 9.0, Graphpad software Inc.).

## Measurements of FRET using acceptor photobleaching microscopy

Cells were grown in deep 24-well plates as described above, and 1 mL of day culture was washed twice and resuspended in 70 μL tethering buffer. The surface of a glass bottom 96-well plate was treated with 0.1% poly-l-lysine for 15 min and rinsed with water twice. The cells were then added to the wells and incubated at room temperature for 10 min to allow attachment. Each well was gently rinsed with tethering buffer twice to remove unattached cells, and incubated with 200 μL tethering buffer afterward.

Measurements of the FRET efficiency by acceptor photobleaching were performed as described previously[86,87]. Briefly, the measurement was conducted using a widefield Eclipse Ti-E inverted fluorescence microscope (Nikon, Japan) equipped with an X-Cite Exacte LED light source, a perfect focus system (PFS), and NIS-Elements AR software (version 4.40, Nikon). Excitation power was adjusted by controlling the fluorescence lamp power and neutral-density filters. Images were

acquired with a 40× air objective, and recorded using iXon 897-X3 EMCCD camera (Andor) with EM gain set to 250. The acquisition time was 1 s. The donor (mTurquoise2) was excited at 436/20 nm and its emission was detected at 480/40 nm. The acceptor (mNeonGreen) was excited at 504/12 nm and its emission was detected at 554/23 nm. The acceptor photobleaching was conducted by a 12-s excitation with 515 nm solid-state laser (CNI, Chuangchun, China). Two acceptor-channel and a hundred donor-channel images were taken before the photobleaching of the acceptor, and forty donor-channel and two acceptor-channel images were taken after the photobleaching. FRET efficiency was calculated as follows:

$$E = \frac{F_{AB} - F_{BB}}{F_{AB}} \times 100\% \tag{2}$$

where background-corrected $F_{AB}$ and $F_{BB}$ were respectively donor fluorescence after and before acceptor photobleaching. Linear fitting of the donor fluorescence intensities versus time was performed for both pre- and postbleaching curves using RStudio to correct for donor photobleaching during data acquisition.

## Ratiometric FRET measurements using microscopy

To measure response kinetics of selected biosensors, $\Delta dgcE$ cells carrying biosensors were grown and then permeabilized using toluene as described above. One milliliter of toluene-permeabilized cells (see above) in the assay buffer (50 mM potassium phosphate [pH 7.4], 100 mM KCl, 0.5 mM MgCl$_2$, 13 mM NaCl) was washed twice and concentrated into a 30 μL volume. A coverslip was treated by 0.1% poly-L-lysine for 15 min and rinsed with water twice. Permeabilized $\Delta dgcE$ cells were incubated on the coated coverslip for 20 min and the coverslip was mounted onto the bottom of a flow chamber.

The ratiometric FRET measurements were performed as described previously[34]. Briefly, the measurement was conducted using a widefield Eclipse Ti-E inverted fluorescence microscope (Nikon, Japan) equipped with an X-Cite Exacte LED light source, a perfect focus system (PFS) and NIS-Elements AR software (version 4.40, Nikon). Cells attached to coverslips at the bottom of the flow chamber were simultaneously imaged in the donor (mTurquoise2, detected at 472/30 nm) channel and the acceptor (mNeonGreen, detected at 554/23 nm) channel using Optosplit (OptoSplit II, CAIRN Research) upon the donor excitation at 436/20 nm. Images were acquired with a 40× air objective, and continuously recorded using iXon 897-X3 EMCCD camera (Andor) with the acquisition time of 1 s. The flow was controlled using a syringe pump (Harvard Apparatus). A stepwise addition and subsequent removal of indicated concentrations of c-di-GMP in the assay buffer was applied at a constant rate of 0.3 mL/min. The region of interest (ROI), fully covered with a monolayer of cells, was selected throughout the entire time series. The ratio between fluorescence emission in the acceptor and the donor channel while exciting donor fluorescence (FRET ratio) was recorded.

## $K_D$ measurements for purified biosensor proteins

Selected biosensor genes from our toolbox were cloned in the pET28 expression vector[88] to create N-terminal fusions with 6×His-tag. These tagged biosensor proteins were overexpressed in E. coli BL21 (DE3) and purified as previously described[88]. Briefly, E. coli BL21 (DE3) cells harboring expression plasmid were grown in Luria-Bertani medium (LB; 1% tryptone, 1% NaCl and 0.5% yeast extract) supplemented with 50 μg/mL kanamycin at 37 °C with shaking at 200 rpm until $OD_{600} \sim 0.6$. Protein expression was induced with 0.1 mM IPTG and cultures were incubated at 16 °C with shaking at 120 rpm for 12 h. Cells were harvested and resuspended in binding buffer (20 mM sodium phosphate, 500 mM NaCl, 20 mM imidazole, pH 7.4) supplemented with lysozyme (0.2 μg/mL), MgCl$_2$ (1 mM), and phenylmethylsulfonyl fluoride (PMSF, 1 mM). Cells were lysed by sonication and centrifuged. The

supernatant was applied to a His GraviTrap column (Cytiva). Proteins were eluted with elution buffer (20 mM sodium phosphate, 500 mM NaCl, 500 mM imidazole, pH 7.4), and dialyzed with the assay buffer (50 mM potassium phosphate [pH 7.4], 100 mM KCl, 0.5 mM MgCl$_2$, 13 mM NaCl). To assess binding to c-di-GMP, the assay buffer containing indicated concentrations of c-di-GMP was added to the purified biosensor protein in a 96-well plate. The final concentration of the biosensor protein was 50 nM. After equilibration at room temperature for 20 min, fluorescence spectra were recorded using a plate reader (Tecan infinite M1000Pro), with excitation at 425 nm and emission scan from 450 to 600 nm at 2-nm interval. The data were acquired with i-control software (version 2.0, Tecan). The fluorescence background obtained by measuring the buffer alone was subtracted from the emission intensity at each interval. FRET ratios at each c-di-GMP concentration were calculated and used to generate dose-response curves.

## Mother machine microfluidics

The mother machine microfluidics device was prepared as described previously[28]. Briefly, the mother machine chip was made with a polydimethylsiloxane (PDMS) mixture (Sylgard 184 PDMS, VwR International GmbH, Germany), which is composed of the curing agent and base in a 1:7 ratio. The liquid PDMS mixture was then poured onto our in-house silicon wafer, placed in a desiccator for 30 min to remove bubbles and then baked at 80 °C overnight. The cured PDMS was peeled off the wafer, resized, and punched for openings. Before imaging, the chip and the glass slide were cleaned using plasma cleaner and subsequently bonded together by baking for 1 min at 80 °C.

Cultures were grown at 37 °C in 10 mL TB containing ampicillin and 5 μM IPTG to an OD$_{600}$ ~ 0.8 and loaded into the mother machine through the inlets. Cells in the mother machine were grown at 37 °C with a constant supply of conditioned medium supplemented with ampicillin, 5 μM IPTG and 0.5 mg/mL bovine serum albumin (BSA) through the inlet at a defined pressure of 200 mbar (LineUP FlowEZ, FLUIGENT, France). The conditioned medium was prepared as described previously[28], by cultivating wildtype cells in TB to OD$_{600}$ ~1.4, removing bacteria by centrifugation at ~3000 g for 10 min and subsequent filtering, and stored at 4 °C. The medium was kept in an ice box throughout the experiment.

The experiment was performed on a widefield Eclipse Ti-E inverted fluorescence microscope (Nikon, Japan) equipped with an X-Cite Exacte LED light source, a perfect focus system (PFS) and NIS-Elements AR software (version 5.20, Nikon). Time-lapse phase-contrast and fluorescence images were captured with a 100× oil objective, and recorded using the Hamamatsu Photonics camera. Upon the donor (mTurquoise2) excitation at 436/20 nm, donor emission was detected at 472/30 nm and the acceptor (mNeonGreen) emission was detected at 542/27 nm. Both phase contrast and FRET images were taken every 10 min. Single-cell segmentation was performed using a previous tool, adapted for phase-contrast images of cells growing in a mother machine[28,89]. The resulting segmented masks were used to quantify fluorescence in individual cells, with single-cell fluorescence defined as the median pixel intensity within each cell outline.

## FRET-To-Sort

To enable FRET-based sorting of mutants according to their c-di-GMP levels, indicated biosensors from our toolbox were transformed into the previously described randomly barcoded Tn5 transposon mutant library[26] by electroporation. After 2 h of recovery at 37 °C, cultures were diluted, plated on LB agar supplemented with ampicillin and kanamycin, and grown overnight at 37 °C. The colonies were washed off the plates using 20% (w/w) glycerol in TB medium, and stored at −80 °C in 500 μL aliquots.

Ten microliters of the glycerol stock of the transformed Tn5 transposon mutant library were used to inoculate the culture grown overnight in 5 mL TB medium supplemented with ampicillin at 37 °C with shaking at 200 rpm. Fifty microliters of overnight culture were diluted into 5 mL fresh medium containing 40 μM IPTG. Cells were incubated for ~4 h at 37 °C with shaking at 200 rpm to OD$_{600}$ ~ 0.4. A hundred microliters of the culture were diluted in 2 mL tethering buffer, briefly vortexed to fully disrupt occasional cell aggregates, equilibrated for 1 h and vortexed again prior to sorting.

Cells were sorted at 4 °C using BD FACSAria™ Fusion cell sorter (BD Biosciences, Germany), with BD FACSDiva software (version v8.0.1, BD) During sorting, fluorescence was excited at 445 nm, and the donor (mTurquoise2) and the acceptor (mNeonGreen; FRET signal) emission was detected through 470/15 nm and 520/35 nm, respectively. Doublets and debris were excluded by gating using FSC and SSC. Cells with, respectively, highest or lowest ratio between signals in the FRET channel and the donor emission channel (1-2% of the population) were selected, as illustrated in Fig. 3a. A tube containing 2 mL of TB medium supplemented with ampicillin was used to collect the sorted cells, with 50 thousand cells per tube. Sorted cells were allowed to recover at 37 °C for 2 h without shaking, and divided into two parts: One milliliter of recovered samples was diluted, plated on LB agar supplemented with ampicillin and kanamycin and incubated overnight at 37 °C. Single colonies were then grown overnight in 5 mL TB, diluted at ratio of 1:100 in 5 mL of fresh TB containing 40 μM IPTG, grown for 4 h at 37 °C with shaking at 200 rpm, resuspended in tethering buffer, and analyzed by flow cytometry for c-di-GMP levels, as described above. The barcodes from colonies showing comparable c-di-GMP levels to Δ$pdeH$ strain or to Δ$dgcE$ strain were amplified by colony PCR using Q5 polymerase (New England Biolabs, Ipswich, Massachusetts) with P1 (AATGATACGGCGACCACCGAG) and P2 (CAAGCAGAA-GACGGCATACGA) primers and characterized by Sanger sequencing. The other one milliliter of the recovered samples was inoculated into 4 mL of fresh TB media supplemented with ampicillin and grown overnight at 37 °C with shaking at 200 rpm. Fifty microliters of overnight culture were grown in 5 mL of fresh TB medium containing 40 μM IPTG before repeating the sorting step. The rest of overnight culture was used to extract genomic DNA using NucleoSpin Microbial DNA kit. Barcode sequences were amplified by PCR using Q5 polymerase with U1 and U2 primers for Next Generation Sequencing (NGS). The PCR reactions were pooled together (10 μL from each reaction), purified using 0.9X of AMPure beads (Beckman Coulter, Brea, California) and sequenced in the GeneCore facility (EMBL, Heidelberg, Germany). The samples were run using the Novaseq 6000 SP Reagent Kit (100 cycles, Illumina, San Diego, California) with 30% PhiX, providing us around 8 million reads per sample.

The barcodes from each sample were extracted using the CutAdapt tool[90], with the sequences U1 (GTCGACCTGCAGCGTACG) and U2 (AGAGACCTCGTGGACATC) as adaptors, a minimum size of 20 bp, and quality higher than 20. The barcodes were then reverse complemented, counted and matched with the Tn5 transposon mutant library map. The read of each gene is the sum of the frequency of barcodes inserted in this gene, which was firstly normalized by total reads of all genes and then by the read of each gene in unsorted sample, to eliminate the difference between transposon library respectively transformed with C3 and D3 biosensors. For mutants enrichment, genes with reads in all samples lower than 200 were excluded from further analysis. The significance of difference in gene enrichment between high and low sorting samples was analyzed using unpaired t-test in Cyber-T web server[91].

## Measurements of curli gene expression

Cells containing a $csgA$ promoter-GFP (green fluorescent protein) fusion (pVS1621)[92] were grown overnight at 30 °C with shaking at 200 rpm in TB medium supplemented with kanamycin in deep 24-well plates to OD$_{600}$ ~ 2.0. Fifty microliters of overnight culture were diluted into 1 mL tethering buffer prior to measurement using BD LSRFortessa SORP cell analyzer (BD Biosciences, Germany) with BD

FACSDiva software (version v8.0.1, BD). Signals were measured with 488 nm laser excitation and 510/20 nm emission. For each sample, the mean fluorescence intensity for a total of 30 thousand events was acquired.

## Measurements of membrane potential
Cultures without fluorescence reporter were grown in TB medium (no IPTG or antibiotics) as described above. Fifty microliters of day culture were diluted into 1 mL tethering buffer supplemented with a membrane potential indicator[61] Thioflavin T (ThT; 596200, Sigma-Aldrich) at a final concentration of 4 µM, and cells were allowed to equilibrate for 1 h prior to measurements. ThT signals were measured using BD LSRFortessa SORP cell analyzer (BD Biosciences, Germany) with BD FACSDiva software (version v8.0.1, BD). Signals were measured with 445 nm laser excitation and 470/15 nm emission. For each sample, the mean fluorescence intensity for a total of 30 thousand events was acquired.

## Measurements of c-di-GMP in cells treated with DNP or antibody
Cells carrying biosensors were grown in TB medium as described above. For DNP treatment, fifty microliters of day culture were diluted into 1 mL tethering buffer with the addition of DNP at a final concentration of 45 µM for 1 h prior to measurements. For FliC antibody treatment, FliC antibody (AA 2-498-FITC, ABIN2831532, 1.5 mg/mL, CUSABIO) was diluted 1:100 into tethering buffer. To reduce the non-specific antibody binding to cell surface, 200 µL of this antibody dilution was incubated for 40 min at room temperature with Δ*fliC E. coli* cells harvested from 20 mL of day culture and subsequently filter-sterilized. One milliliter of day culture was washed and resuspended in 1 mL tethering buffer. Ten microliters of cell suspension were incubated with 10 µL of filtered antibody solution or control tethering buffer for 20 min, and diluted into 200 µL of tethering buffer prior to measurements. FRET measurements of c-di-GMP levels were performed using flow cytometry as described above.

## Measurement of motility upon antibody treatment
Antibody-treated samples were prepared as described above. Ten microliters of cell suspension were incubated with 10 µL of filtered antibody solution or tethering buffer for 20 min. 2 µL droplet of this mixture was then sandwiched between two coverslips using a 140-µm thick spacer. The chamber was sealed with grease to prevent evaporation. Motility measurement was conducted as previously described[93,94]. Briefly, movies were taken at the center of the droplet (away from surfaces) using a Nikon TI Eclipse phase-contrast microscope (Nikon, Japan) equipped with an EoSens 4CXP CMOS camera, StreamPix 6 software (version multi-camera, NorPix) and a 10× air objective. Images were acquired with an exposure time of 1 ms at a rate of 100 frames per second (fps) for 100 s, with 2 × 2-binned pixels of 1.4 µm. The fraction of swimming cells was determined by analyzing these movies using Differential Dynamic Microscopy (DDM)[94,95]. To visualize the swimming trajectories of cells, another movie was taken at a rate of 50 fps for 40 s using the same camera with non-binned pixels of 0.7 µm, and analyzed using the Particle_Tracking_2 ImageJ Plugin[93,96].

## Measurements of c-di-GMP and motility at high viscosity
Cells were grown in TB medium as described above. For c-di-GMP measurements, one milliliter of day culture was washed and resuspended in 200 µL tethering buffer. 180 µL of bacterial suspension were mixed with 20 µL tethering buffer with 20% (w/v) ficoll, or without ficoll, and transferred to a 96-well plate for imaging. C-di-GMP levels were quantified by measuring the FRET ratio using the same microscopy setup described above for ratiometric FRET measurements. For motility measurements, 1 mL of day culture was washed and resuspended in 1 mL motility buffer (20 mM potassium phosphate, 0.1 mM EDTA, 0.01% Tween-80, pH 7), and 100 µL of this suspension was mixed with 100 µL motility buffer with 4% (w/v) ficoll. Sample preparation, image acquisition and DDM data analysis for motility measurements were performed as described above.

## Measurement of c-di-GMP during surface colonization
Cells carrying biosensors were grown in TB medium overnight as described above. One hundred microliters of overnight culture were diluted into 10 mL fresh medium incubated at 37 °C with shaking at 200 rpm to $OD_{600}$ ~ 0.35. The samples were further diluted to $OD_{600}$ ~ 0.04 in fresh TB medium supplemented with 40 µM IPTG and ampicillin, and 400 µL of this suspension was seeded per well into the 8-well microscope slides with glass bottom (µ-Slide 8 well glass bottom, ibidi GmbH, Germany). Cells were incubated at 37 °C statically for 3 h and subsequently imaged using confocal microscopy.

Cells were imaged using an inverted Zeiss Axio Observer Laser Scanning Microscope (LSM) 880 equipped with a 40× water objective and Zeiss Zen software (version 2.1 SP3, Zeiss). Images were acquired with excitation by a 440 nm laser and emission at 446-499 nm for mTurquoise2 and at 526-579 nm for mNeonGreen. For each field of view, images were acquired at different z-positions, focusing either on single cells attached to the surface or on cell aggregates above the surface. For Lambda-mode spectral imaging, the light path was switched from Channel Mode to Lambda Mode, and the detection range was 446-695 nm in 14 channels with 17.8 nm spectral resolution. Images using Lambda Mode were exported using Zeiss ZEN lite software (version 3.13, Zeiss). C-di-GMP levels were quantified by determining the FRET ratio.

## Statistics and reproducibility
Data statistics were analyzed using SPSS Statistics (Version 29.0.2.0, IBM), Microsoft Excel 2021 (Microsoft), and GraphPad Prism (Version 9.0, Graphpad software Inc.), and statistics of NGS data were analyzed in the Cyber-T web server[91]. Details of statistical analyses are provided and exact *P* values are provided in the Source Data file. No statistical methods were used to predetermine the sample size. Generally, for statistical analysis, at least three biological replicates were performed, except for the FRET-To-Sort enrichment that was performed in two biological replicates because of the experimental complexity. No data were excluded from the analyses. The experiments were not randomized.

## Reporting summary
Further information on research design is available in the Nature Portfolio Reporting Summary linked to this article.

## Data availability
NGS datasets used in this study are available in the NCBI database under accession BioProject PRJNA1144054 [http://www.ncbi.nlm.nih.gov/bioproject/1144054]. Flow cytometry and microscopy raw data are available from [https://doi.org/10.17617/3.BFOX3H][97]. The data presented in the paper and the Supplementary Information are available in the Source Data file. Source data are provided with this paper. The biological materials are available upon request under a Material Transfer Agreement. Source data are provided with this paper.

## Code availability
Code of DDM is publicly available from [https://doi.org/10.5281/zenodo.3516258][95]. Code of Particle_Tracking_2 ImageJ Plugin is publicly available from [https://doi.org/10.5281/zenodo.18890231][96]. Code used for cell segmentation in mother machine experiments is available from https://git.ecdf.ed.ac.uk/swain-lab/baby.

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

## Acknowledgements

This research was funded by the Max-Planck-Gesellschaft (to V.S.) and by National Institutes of Health Grant R01 GM124589 (to M.H.). We thank Dr. Remy Colin for valuable discussions and help with the data analysis.

## Author contributions

L.W., G.M. and V.S. designed research; L.W., G.M., A.S.M., X.C. and S.G.S. performed experiments; L.W., G.M., A.S.M., X.C. and J.P. analyzed data; M.C.H. and N.T. contributed materials and L.W. and V.S. wrote the paper.

## Funding

## Competing interests

The authors declare no competing interests.
