## [Transparent Peer Review file · Nature Communications]

Toolbox of FRET-based c-di-GMP biosensors and its FRET-To-Sort application for genome-wide mapping of c-di-GMP regulation

Corresponding Author: Professor Victor Sourjik

Version 0:

Reviewer comments:

Reviewer #1

(Remarks to the Author)

The manuscript by Wang et al. describes the development of a set of FRET-based biosensors for the bacterial second messenger c-di-GMP. The design is based on the c-di-GMP binding PilZ-domain of the YcgR protein, which has previously been used to construct similar sensors (PMID: 20522779, PMID: 23163901). By using different homologs of YcgR established earlier for BRET-based c-di-GMP sensors in vitro (PMID: 29466657), the authors assemble a set of sensors with c-di-GMP affinities ranging over two orders of magnitude. They then combine two of their FRET sensors with a Tn-Seq/FACS screen to identify genes influencing c-di-GMP levels in *E. coli*.

Major points:

Because real-time biosensors are invaluable tools to study dynamics of metabolites in vitro and in vivo, this study is relevant. Unfortunately, the authors fail to demonstrate how their sensors perform compared to existing FRET-based biosensors with essentially the same design (PMID: 20522779, PMID: 23163901). This should include in vivo studies tracking characteristic cell-to-cell variations of c-di-GMP in any of the different model systems available, as well as in vitro analyses with purified sensors to determine excitation and emission spectra and dose-response curves. The methodology to derive KDs and kinetics using permeabilized cells is certainly valid as an initial screening approach but does not substitute for an in-depth in vitro characterization of the sensors' kinetic properties.

The authors make the valid argument that biosensors with different affinities are required to measure varying cellular concentrations of c-di-GMP in vivo and in individual cells. One would expect the authors to showcase the suitability of their biosensors to substantiate this claim by actually testing them in organisms with different physiological regimes of the ligand. This seems particularly relevant as PilZ-based FRET sensors with a >100-fold range of affinities for c-di-GMP already exist (PMID: 20522779, PMID: 23163901).

On a more technical note, flow cytometry data are plotted as 'mean +/- SD', which only take peak values into account, while ignoring distributions. Population distributions (e.g., histograms of single cell data) should be shown directly in the main figures or in the supplementary or source data to allow the reader to judge how well their sensors separate populations with different c-di-GMP concentrations. In fact, dot plots in Fig. 1e,f give the impression that c-di-GMP high and low populations for both D3 and F8 biosensors largely overlap and are poorly separated. Similarly, the dose response curves in Fig. 2b and Extended Data Fig. 2 show large error bars even for mean values, which would imply that population distributions give an even noisier picture.

Overall, the novelty of this work lies in the development of improved FRET sensors and not necessarily in the combination of FRET sensors with FACS as highlighted in the title (similar screens have been published before; PMID: 30395379, PMID: 26060330). Given that similar FRET-based sensors (PMID: 20522779, PMID: 23163901) and other biosensors for c-di-GMP (PMID: 34961989, PMID: 38724508) already exist, it is important to demonstrate that the new sensors are superior and offer added value to existing tools.

Additional comments:

- Please provide single event data for all flow cytometry (and microscopy) measurements, e.g., as histograms in supplementary files.
- Please include statistics. E.g., Supp Fig 2: most biosensors do not seem to differ between *pdeH* and *dgcE* strains, but they state in the main text that this figure served to exclude biosensors that did not respond. Yet B1, F2, E2, C1 and F8, which show no difference in this Figure, were still included in Fig 2b and others.
- The observation that ionic strength influences c-di-GMP levels is interesting and is certainly worth mentioning in the manuscript, but given that the underlying mechanism/cause is not addressed, does not seem to merit an entire paragraph.
- The genetic screen is only peripherally related to the main topic of the manuscript and fails to showcase the potential strengths of a FRET sensor. In fact, it is not clear why a FRET-based reporter would be required for such a screen at all. A simple transcriptional reporter would do the same trick. Also, the authors use both C3 (medium affinity) and D3 (high affinity) sensors in their screens (with different salt concentrations to achieve different basal c-di-GMP levels) and find 'similar results' (l. 203). Why did the authors perform the experiment in this way (different salt concentrations), rather than using the same conditions and see if they, in fact, get different hits depending on the KD of the sensor? This undermines the narrative that biosensors with different affinities are required for such studies.
- One would have expected that in their genetic screen for c-di-GMP high mutants, the authors identify *pdeH* itself. Is there an explanation for why this was not the case?

Reviewer #2

(Remarks to the Author)

Wang et al. have modified a previously published 92-member chemiluminescent YcgR biosensor toolkit (BRET, Reference 28), to a FRET-based one, because the former did not work as well in Gram negative bacteria. They characterized and selected a subset of these biosensors with different affinities for c-di-GMP, and used these to (1) test the response of the highly active DgcE to NaCl and 2) transform them into a random-barcoded Tn5 mutant library to identify mutants with low or high c-di-GMP determined by FACS sorting. While the specific experiments with DgcE conclude that its activity is responsive to ionic but not to non-ionic osmolytes, the data are not straightforward and raise more questions than they answer as noted below. The FRET-to-SORT part of this work is methodology development. It identified genes, which when mutated, show high or low c-di-GMP levels. The majority of hits that show increased c-di-GMP are in the well-studied flagellar regulon, which is not surprising because *pdeH* is a late flagellar gene and therefore will not be expressed in the absence of early flagellar genes. However, mutations that lower c-di-GMP map to flagellar genes *motA*, *motB* and *fliC* that are not expected to interfere with *pdeH* expression. Why this is so is not investigated. In other words, no significant new insight into the c-di-GMP network in *E. coli* has emerged from use of this toolbox.

Major comments related to DgcE being a sensor of ionic osmolytes

1. Fig. 3c presents c-di-GMP levels as measured by LC-MS/MS, which indicate that these levels are sensitive to NaCl in WT and *pdeH*. The *dgcE* mutant is unresponsive and has 5000 times less c-di-GMP compared to *pdeH*, implying that DgcE is contributing to practically ALL the c-di-GMP in the cell. Missing in this analysis is important data for the *pdeH dgcE* double mutant, because subsequent experiments in Fig. 3e-g all use the double mutant (orange line) and show substantially high levels of c-di-GMP i.e. only 10% lower than WT. So, there appears to be a major discrepancy between what the biosensors measure and the actual concentrations of c-di-GMP.
2. Fig. 3e,f. The authors claim that c-di-GMP levels are not affected by NaCl or Kglutamate in the *pdeH dgcE* mutant (orange line) compared to WT even though the data looks exactly like WT in which these levels ARE affected.
3. The conclusion that DgcE is responsive to ionic osmolytes is premature in that the authors are monitoring c-di-GMP levels in the complete absence of this DGC in all their experiments. If DgcE associates with partners in the membrane, the effect could be indirect. It would have been more appropriate to use an active site mutant instead.
4. The authors invoke depolarization of the membrane and perturbation of membrane potential to explain the effect of NaCl on DgcE activity, but no actual measurements are made for the only 'new' function they claim to have uncovered in this paper.

Other comments

Although 18 sensors were selected for their sensitivity, only two are used in all the analyses without much explanation.

Line 38: While it is true that transcriptional sensors may be sensitive to other cellular factors, FRET-based approaches are not immune to such influences either.

Lines 46-47: This recent publication suggests otherwise:
<https://www.nature.com/articles/s41467-024-48295-0>

Lines 78-81: The cited review does not propose that DgcE is a major DGC contributing to the global pool of c-di-GMP. Previous studies have shown that deleting DgcE does not significantly affect overall c-di-GMP levels, although it does influence biofilm production (PMID: 29018125). The authors may want to address and clarify the discrepancy between their findings and these earlier results.

Line 162: In Fig. 3B, why are the data plots for NaCl concentration at 60mM absent, when this is approximately the concentration used in their buffers? In Extended Fig. 3 it is unclear what the different NaCl concentrations refer to.

Line 170: What is the rationale for switching the sensor A3 in Fig 3d to C3 in Fig. 3e-g? Based on previous data, both are medium-affinity biosensors.

Line 201: Fig. 3a shows no response of the D3 sensor to 67mM NaCl. Why then is 134mM NaCl used in sorting experiment with this sensor? Is it because it lowers c-di-GMP levels even further?

Line 209: If the low-affinity biosensor is more sensitive to decreases in c-di-GMP levels (line 167), why is the high affinity D3 sensor being used to screen for cells with low c-di-GMP levels as shown in Fig. 4a?

Line 230: BluF is involved in biofilm maturation, but it is not related to c-di-GMP production; BluF has a degenerate EAL domain.

Line 243: Extended Fig. 7a. If these transposon mutants were selected based on FRET signal higher than pdeH, why are all the mutants showing a lower signal compared to pdeH?

Reviewer #3

(Remarks to the Author)

Reviewer #4

(Remarks to the Author)

Version 1:

Reviewer comments:

Reviewer #4

(Remarks to the Author)

Summary:

The manuscript by Wang et al. reports the development of a FRET-based biosensor kit capable of detecting a broad range of c-di-GMP concentrations in real time, spanning from 9 nM to 1 μ M. The study screened 16 distinct sensor variants, each exhibiting different affinities for c-di-GMP, thereby enabling researchers to select sensors best suited for specific experimental conditions or bacterial species of interest. Using a microfluidic platform, the authors monitored live cells harboring the biosensors and observed remarkable cell-to-cell heterogeneity in intracellular c-di-GMP levels. They further integrated the FRET-based sensors with fluorescence-activated cell sorting (FACS) to perform high-throughput screening of a *E. coli* Tn5 transposon mutant library, identifying mutants with either elevated or reduced c-di-GMP levels relative to Δ dgcE and Δ pdeH reference strains. Interestingly, the screening revealed opposing effects of mutations in flagellar class II versus class III genes: disruptions in class II genes led to an increase in c-di-GMP levels, whereas class III mutations caused a decrease. The authors further linked the reduction in c-di-GMP levels to an increase in membrane potential, suggesting a mechanistic connection between flagellar assembly defects and c-di-GMP signaling dynamics.

The authors have addressed most of the concerns raised in the previous version of the manuscript and provide new insights into the potential relationship between membrane potential and c-di-GMP dynamics in *E. coli*. The manuscript could be further strengthened and clarified by addressing the comments outlined below.

Previous comments:

1. All comments regarding osmolytes are no longer relevant, as the authors have decided to remove this section entirely and focus instead on validating the sensor and explaining changes in c-di-GMP levels resulting from mutations in flagellar class II or III genes.
2. The justification for using different sensors in each assay has been provided and is now easier to understand.
3. Minor textual revisions have been made, improving overall clarity.

Comments on the Revised Manuscript:

1. In line 107, the authors previously stated that FRET efficiency provides a more accurate readout than the FRET ratio, as it is less influenced by bleed-through and biosensor expression levels, which makes the FRET ratio less suitable for live-cell tracking. However, in Supplementary Figure 7 (Figure S7), the authors use the FRET ratio to measure c-di-GMP levels in

live cells. While this choice is understandable given potential technical limitations, the earlier emphasis on FRET efficiency as the better metric raises concerns about the robustness of the assay.

2. The authors use Figure S7 to support their conclusion that the sensor can monitor c-di-GMP levels in live cells and that “high dynamics and heterogeneity of c-di-GMP levels in growing *E. coli* cells, which were not apparently correlated with the cell cycle” (line 167). However, when comparing the performance of the F1 sensor under two conditions—Figure 2d and Figure S7a—it remains unclear:

- a) whether comparable output levels should be expected for equivalent c-di-GMP concentrations, and
- b) if so, whether the data in Figure S7a imply that intracellular c-di-GMP levels among individual cells range from approximately 10 nM to 810 nM.

Although prior studies have reported heterogeneity in c-di-GMP concentrations, it is uncertain whether the present data substantiate such a wide dynamic range. It would strengthen the manuscript if the authors could further discuss the expected magnitude of these dynamics based on their data and how their observations compare with previously reported ranges.

3. In both Figure 2b and Supplementary Figure 4a, the 16 FRET sensors exhibit the dose-dependent responses described by the authors. While most sensors show relatively low variability within the concentration range suitable for measuring c-di-GMP levels in wild-type *E. coli* cells, the substantial standard deviations observed at higher concentrations (approximately 100–1000 nM) make it difficult to determine whether most of these sensors are also reliable for detecting elevated c-di-GMP levels in bacteria such as *P. aeruginosa*.

4. In line 286 and Figure 4D–E, DNP was used to dissipate the proton motive force (PMF) in wild-type cells, resulting in an observed increase in c-di-GMP levels. It is unclear, however, whether the same treatment was applied to the *motA* and *fliC* deletion mutants. Based on Figure 4D, where DNP collapses most of the PMF, one would expect these mutants to exhibit c-di-GMP levels comparable to those of the DNP-treated wild type, despite their inherently elevated membrane potential. Including this control would strengthen the interpretation that the increased membrane potential in these mutants contributes to the observed decrease in c-di-GMP levels.

5. In line 299, are the authors suggesting that c-di-GMP levels decrease upon surface contact as a consequence of increased membrane potential? It would be important to clarify whether the antibody concentration used in the experiment fully inhibits motility, as this detail is crucial for interpreting the proposed mechanism. The prevailing understanding is that c-di-GMP levels generally rise upon surface contact, although the timing and extent of this increase can vary. As mentioned in line 281, stator engagement is load-dependent; however, it remains unclear whether antibody binding is perceived by the cells as an increased mechanical load, potentially triggering additional stator recruitment. Moreover, do the authors observe a similar decrease in c-di-GMP levels when cells experience increasing gradients of viscosity? Please clarify lines 300–302, which currently suggest that surface contact leads to reduced c-di-GMP production, or expand the discussion to reconcile this observation with existing models.

Minor comment

1. Given that the toolkit contains 16 sensors, what would be the most straightforward approach to determine which sensor is best suited for different bacterial species or experimental conditions? Can the dissociation constants and FRET efficiencies characterized in *E. coli* across various c-di-GMP concentrations serve as an initial reference point, or would users need to conduct screening experiments similar to those performed by the authors to identify the optimal sensor? As this study introduces a new experimental tool, it would be helpful—though not strictly necessary—to include practical guidance or recommendations for sensor selection.

Reviewer #5

(Remarks to the Author)

In the manuscript “A toolbox of FRET-based c-di-GMP biosensors and its FRET-To-Sort application 1 for genome-wide mapping of the second messenger regulatory network,” Wang et al describe the production of a set of FRET biosensors for the secondary messenger cyclic-di-GMP and its use through a FACS-based transposon mutagenesis screen. This screen identified two sets of flagellar mutants that affected c-di-GMP levels: one set which increased c-di-GMP in a previously identified PdeH-based mechanism, and a second set that decreased c-di-GMP. The authors investigated the latter and found some evidence to suggest that this was related to disruption of membrane potential through an inactivation of the flagellar machinery. I was asked to serve as a secondary reviewer for this manuscript as the initial Reviewer #1 was unable to review this resubmitted version. I am therefore primarily focusing on the response to the previous reviews and the new material regarding membrane potential regulating c-di-GMP concentrations in flagellar mutants, as I have not read or reviewed the prior manuscript submission containing the NaCl data.

In regards to the response to the prior reviews, the authors have done a good job clarifying where biosensors are being used, and their purification and *in vitro* testing of the biosensor Kds alongside the permeabilization assay has supported their prior findings, which are both positive. However, there are still several concerns with these responses and with the new data on membrane potential regulation of c-di-GMP:

- The authors mention that one novelty of this work is the production of a biosensor library with a range of binding affinities. In PMID: 23163901 mentioned by the prior first reviewer, a biosensor library consisting of Kds ranging from 88nM to 8.24μM were produced, covering roughly a range of 27nM to 26μM using their one-log coverage range. So while the authors are correct that their especially high affinity biosensors are new to the biosensor field, the concept of using a range of binding

affinities is established.

- While there are other publications that use flow cytometry to measure biosensors, I agree that incorporating a sorting and Tn-Seq based analysis is a useful tool for further study. Since the sorting is the novel component though, I'd like to see a little more data on how effectively mutants were sorted. Higher c-di-GMP bacteria especially may be more prone to clumping together (PMID: 26060330), which could cause a subsequent mutant that was similarly sticky to be brought along during sorting. From my reading, it sounds like the authors sorted for c-di-GMP and subjected a partial mixture to individual colony isolation. Clones that showed either high or low c-di-GMP were Sanger sequenced, but how efficient was isolation? In other words, how many sorted/isolated colonies no longer showed altered c-di-GMP that were excluded from further testing, but that might appear in the NGS sample? Along the same lines, the high c-di-GMP batch appeared to be moderately efficient, with about a third of the NGS-derived clones also being isolated through Sanger sequencing, but there were also clones that were only isolated through Sanger or NGS but not both. This was much more pronounced with the low c-di-GMP set, in which only 10% of indicated mutants were isolated through Sanger sequencing. Is this just a case of a small sample size in Sanger failing to quantify the full mutant set? Or is the sorting accidentally pulling out mutants that appear in NGS data but that were excluded from Sanger sequencing because their c-di-GMP levels don't actually change? Why also is the Sanger sequencing identifying mutants that weren't identified in the NGS sample? Are these Sanger-identified mutants closer to WT c-di-GMP to the point where they didn't make the NGS cutoff? Considering this low data set is their novel high affinity biosensor and the sorting is the novel component for this technique, the authors should comment on these apparent discrepancies and limitations.

- In Figure 4, the authors investigate the role of class II and class III flagellar mutations on c-di-GMP. The class II mutant model makes sense and is well supported by prior work, where a failure to secrete FlgM resulting in low expression of PdeH and high c-di-GMP. The authors hypothesize that the low levels of c-di-GMP seen in class III mutants may be due to over-secretion of FlgM, but test that using a flgM mutant. Wouldn't the flgM mutant just mimic FlgM over-secretion? For instance, if the fliC mutation is causing a higher than typical FlgM secretion and deleting flgM completely lowers c-di-GMP even further, it would seem that would provide evidence to the FlgM over-expression hypothesis. I don't think these experiments show a FlgM-independent effect for the type III mutants as the authors claim, at least based on this data.

- In Fig. 4c, the authors investigate this further, but have to use a pdeH mutant in order to move c-di-GMP levels into a range detectable by their biosensors, claiming that the further reduction of c-di-GMP beyond the fliC mutant level may not be possible with their high affinity (D3) biosensor. Were these experiments attempted with the D3 biosensor? Considering these high affinity biosensors are the one novel set of biosensors, this would seem to be an excellent test case for their utility. One potential downside to measuring these effects at higher concentrations is that you may be reaching the DGC I-site binding affinity for DgcE, causing it to repress independent of the desired response. Is there a predicted level of c-di-GMP at these various biosensor levels, and is the I-site K_d for DgcE known?

- In Figure 4g, the authors see a small uptick in FRET efficiency upon treatment with DNP in a DgcE-dependent manner. However, they're using a low affinity F2 biosensor in this figure that is displaying a FRET efficiency of around 24%. In Supplementary Figure 3, their F2 biosensor appears to bottom out in the dgcE mutant at 30%, ranging from 30-45%. Are these FRET efficiencies calculated differently between figures? Is it possible that the biosensor is unable to detect c-di-GMP concentrations below the point at which the authors are measuring? There's no low c-di-GMP control to show that these unchanged values aren't due to the fact that they are just outside the range of biosensor measurement. As a large point of the manuscript is the utility of these biosensors to detect ranges accurately, low and high c-di-GMP mutants (pdeH and dgcE) should be included in every figure to ensure the measurements are within that particular biosensor's range.

- In Fig 4h, the authors use a flagellar antibody to attempt to halt rotation of the flagellum in an effort to indicate this alters membrane potential to inactivate DgcE. Have the authors confirmed these wildtype bacteria are motile at this concentration? According to Fig. 4a, the C3 biosensor appears to max out at a FRET efficiency of 37%, pretty similar to what is seen in the wildtype sample here. The C3 biosensor would apparently measure up to around 230nm, which at the higher end would be sufficient to activate the YcgR flagellar brake (K_d 141nM - PMID: 31836667). How does YcgR fit into this model? Wouldn't YcgR inhibition of flagellar rotation also then cause increased membrane polarization, inactivation of DgcE, reduction of c-di-GMP through PdeH, and subsequent release of the flagellar body by YcgR? This would seem to make YcgR fairly useless as a flagellar brake. The authors also mention that this antibody-based flagellar jamming could mimic surface contact, but don't describe why Salmonella would want to lower c-di-GMP upon surface contact. Wouldn't this be better served by high c-di-GMP and biofilm formation?

- The authors mention (line 337-338) that "perturbation of the cell envelope and membrane composition might generally affect c-di-GMP signaling." They reference this in regard to both low and high mutants, but in their model wouldn't these perturbations universally lead to lower c-di-GMP due to membrane polarization-induced DgcE inactivation? Without a mechanism, my assumption was that high membrane potential somehow inactivated DgcE that would normally be synthesizing c-di-GMP. Or is the suggestion that DgcE is specifically also activated by low membrane potential beyond wild type potential levels? That would be an interesting finding, but doesn't appear to be discussed here. Absent a further biochemical examination of DgcE, the indication that it somehow responds to membrane potential needs better evidence.

- The other issue with the author's model is during class II flagellar mutant analysis. The beginning of the manuscript details how DgcE is the primary cyclase present in E. coli, and that reduction of PdeH levels results in DgcE-mediated c-di-GMP concentrations increasing. According to their model though, these class II mutations should also be disabling DgcE via increased membrane potential. In this situation, what cyclase is producing c-di-GMP in the class II mutants when both PdeH and DgcE are absent? If they test a fliG/dgcE double mutant, are c-di-GMP similarly as high as the fliG mutant? Or would

these levels decrease because DgcE is still active despite the higher membrane potential? None of these double mutants were tested, and the class II mutants weren't tested for membrane potential to provide support for their DgcE membrane potential sensor activity.

In summary, I think there are some useful tools in here. The high affinity biosensors could provide new avenues of study, although it is a little worrying the authors didn't capitalize on this when it was potentially useful. The FRET-to-SORT is also a potentially powerful tool with a little more data to indicate the efficiency of sorting and discrepancies between the Sanger and NGS datasets. The model of membrane polarization affecting DgcE and c-di-GMP levels is the truly novel part here, and there the work is still pretty preliminary. Inclusion of some more controls to support their model would be necessary, and potentially some biochemical mechanisms underlying DgcE detection of membrane potential. I don't know that the biosensor/sorting techniques are novel enough on their own without the membrane potential finding, but strengthening this latter set of findings would greatly improve the manuscript.

-Erik Petersen, East Tennessee State University

Version 2:

Reviewer comments:

Reviewer #4

(Remarks to the Author)

The manuscript is well written, and the authors have addressed my previous concerns satisfactorily.

I am particularly intrigued by the findings regarding regulation of c-di-GMP through the proton motive force (PMF) (Fig. 4D-E). Contrary to the widely accepted model in which inhibition of flagellar rotation leads to increased c-di-GMP levels and subsequent biofilm formation, the authors report that hindering motor rotation results in reduced intracellular c-di-GMP. These findings suggest a more nuanced interplay between flagellar rotation and c-di-GMP regulation. Specifically, the data imply the existence of a "Goldilocks" zone, in which an optimal level of mechanical load promotes maximal c-di-GMP production, as mentioned by the authors (lines 418–423). It would be valuable to see how this model could be integrated into the current framework of surface sensing and biofilm regulation.

Minor comments

1. As mentioned by the authors (line 386), cell envelope stress can affect c-di-GMP levels. It would be interesting if some of the observed phenotypes might actually be linked to alterations in PMF arising from membrane perturbation.
2. The identification of genes not previously associated with the c-di-GMP network is also intriguing. This finding opens new avenues for understanding the architecture of c-di-GMP signaling in *E. coli*, particularly in determining whether these genes function as direct regulators or act indirectly through other pathways.

Reviewer #5

(Remarks to the Author)

In the resubmitted manuscript "A toolbox of FRET-based c-di-GMP biosensors and its FRET-To-Sort application 1 for genome-wide mapping of the second messenger regulatory network," Wang et al describe the production of a set of FRET biosensors for the secondary messenger cyclic-di-GMP and its use through a FACS-based transposon mutagenesis screen. This screen identified two sets of flagellar mutants that affected c-di-GMP levels: one set which increased c-di-GMP in a previously identified PdeH-based mechanism, and a second set that decreased c-di-GMP. The authors investigated the latter and found some evidence to suggest that this was related to disruption of membrane potential through an inactivation of the flagellar machinery.

This is the third submission of this manuscript, and many of the prior questions have been answered to my satisfaction. There remain just a couple of points that require clarification:

- Regarding the Flow-to-Sort experiment, the authors have done a good job clarifying some of the discrepancies between the Sanger and NGS sequencing outputs. As to the potential for clumping in the flow sorting assay, I also appreciate that the authors were able to include forward scatter plots for their various populations that were tested. However, it would be useful to show that there was no clumping in a high c-di-GMP mutant (ie $\Delta pdeH$). It appears that the authors have taken their protocol into account to hopefully minimize these issues, and I imagine that it would work for the majority of the mutants in their collection. It would be those few high c-di-GMP mutants that would become the issue though, so confirming that their established protocol was able to eliminate any clumping in the $\Delta pdeH$ mutant would strengthen their protocol.
- There was also a miscommunication on the issue of the D3 biosensor used for testing in Fig. 4C. I'd previously written "In Fig. 4c, the authors investigate this further, but have to use a pdeH mutant in order to move c-di-GMP levels into a range detectable by their biosensors, claiming that the further reduction of c-di-GMP beyond the fliC mutant level may not be possible with their high affinity (D3) biosensor." My point there was whether the authors had attempted to use the D3 biosensor in the presence of pdeH to investigate these changes. In other words, have they tried to test $\Delta motA$, $\Delta fliC$, Δdgc ,

etc mutants in a pdeH-competent background. If the argument for having to make all of these mutants within a Δ pdeH background was to raise the c-di-GMP high enough to reach the C3 biosensor, why weren't these mutants constructed in a wild type background where PdeH is still present and tested using the D3 biosensor? Is a pdeH-competent background still too low in c-di-GMP to even detect changes using D3? I appreciate that the authors attempted the D3 in the Δ pdeH background in Supp Fig 9, but I was discussing the alternative experiment (D3 in the wild type background where it should have greater utility to detect changes in the low levels of c-di-GMP present in a pdeH-competent strain).

Otherwise, I think the authors have done a good job responding to the rest of my concerns, and I look forward to where this story leads in the future.

-Erik Petersen, East Tennessee State University

Reviewers' comments:

Reviewer #1 (Remarks to the Author):

The manuscript by Wang et al. describes the development of a set of FRET-based biosensors for the bacterial second messenger c-di-GMP. The design is based on the c-di-GMP binding PilZ-domain of the YcgR protein, which has previously been used to construct similar sensors (PMID: 20522779, PMID: 23163901). By using different homologs of YcgR established earlier for BRET-based c-di-GMP sensors in vitro (PMID: 29466657), the authors assemble a set of sensors with c-di-GMP affinities ranging over two orders of magnitude. They then combine two of their FRET sensors with a Tn-Seq/FACS screen to identify genes influencing c-di-GMP levels in *E. coli*.

Major points:

Because real-time biosensors are invaluable tools to study dynamics of metabolites in vitro and in vivo, this study is relevant. Unfortunately, the authors fail to demonstrate how their sensors perform compared to existing FRET-based biosensors with essentially the same design (PMID: 20522779, PMID: 23163901).

Response: We thank the Reviewer for appreciating the importance of developing real-time biosensors and for acknowledging the relevance of our work. We also appreciate the critical points raised by this and other reviewers, which helped us to improve the presentation of our work and to better highlight its novelty.

We have now revised the introduction and discussion of the manuscript to compare our sensors in greater detail with the already available real-time c-di-GMP biosensors, including FRET-based biosensors. In brief, current biosensors offer only a limited choice of affinities, which do not cover the full physiological range of c-di-GMP levels in bacteria, particularly in the low concentration range. Moreover, since each individual sensor can only sensitively detect a narrow, typically one decade, range of c-di-GMP concentrations, and the designs of several available biosensors are different, combining them to assess different c-di-GMP regimes in the same specie (as done in our study) is difficult. Therefore, our aim was to create a toolbox of biosensors with a uniform design and consistent readout, undergoing large FRET signal change upon c-di-GMP binding and displaying a wide and stepwise coverage of c-di-GMP concentrations, including low-concentration regimes that were not previously accessible.

Moreover, for reference, we have now displayed the results for the C1 biosensor, which is based on the same YcgR (STM 1798) from *Salmonella typhimurium* as the previously published FRET biosensor in PMID: 20522779 and PMID: 23163901 (but using mNeonGreen and mTurquoise2 as the FRET pair). In comparison to C1 biosensor, the biosensors in our toolbox showed similar or larger signal changes upon c-di-GMP binding (see Fig.1d,e, Fig.2a and Supplementary Fig.5d), and many had higher affinities (see Table 1).

This should include in vivo studies tracking characteristic cell-to-cell variations of c-di-GMP in any of the different model systems available, as well as in vitro analyses with purified sensors to determine excitation and emission spectra and dose-response curves.

Response: Although the application of our biosensors for the in-depth analysis of c-di-GMP dynamics in growing bacterial cells will be focus of our subsequent work, we agree with the Reviewer that demonstrating their applicability for such studies would strengthen our manuscript. We thus now present new proof-of-principle data on tracking c-di-GMP levels in growing *E. coli* cells at single-cell resolution, using a mother machine microfluidics device. This experiment revealed previously unreported high and dynamic variability of c-di-GMP levels in *E. coli*, which is different from the cell-cycle dependent c-di-GMP dynamics previously observed in other bacteria (see Supplementary Fig.7). To illustrate the advantages

of having biosensors with different affinities, we have performed this experiment with two biosensors from our toolbox, showing that a high-affinity FRET biosensor is more efficient in resolving c-di-GMP heterogeneity.

The methodology to derive K_D s and kinetics using permeabilized cells is certainly valid as an initial screening approach but does not substitute for an in-depth in vitro characterization of the sensors' kinetic properties.

Response: To validate our approach of analyzing sensor affinities using permeabilized cells, we have now performed *in vitro* characterization of c-di-GMP binding for several purified biosensors (see Supplementary Fig.5 and Table 1). The values of K_D obtained using purified biosensors showed a very good quantitative agreement with those derived from experiments using permeabilized cells.

The authors make the valid argument that biosensors with different affinities are required to measure varying cellular concentrations of c-di-GMP in vivo and in individual cells. One would expect the authors to showcase the suitability of their biosensors to substantiate this claim by actually testing them in organisms with different physiological regimes of the ligand. This seems particularly relevant as PilZ-based FRET sensors with a >100-fold range of affinities for c-di-GMP already exist (PMID: 20522779, PMID: 23163901).

Response: Although we agree with the Reviewer that previously published biosensors show a >100-fold difference of affinities for c-di-GMP. However, since each individual sensor can only sensitively respond to a narrow range of c-di-GMP levels, investigation of a wide range of cellular c-di-GMP concentrations requires a large number of sensors with the stepwise coverage of the concentration range. Even more importantly, these existing biosensors are limited to relatively low affinities and cannot resolve c-di-GMP changes in the low-concentration regimes.

We now better illustrate the importance of having different-affinity biosensors by applying those with high and medium affinity in our FRET-To-Sort protocol and in subsequent characterization of the impact of motility-related mutations on c-di-GMP levels in high or low c-di-GMP regimes. We believe that this is sufficient to substantiate our claim that different affinities of biosensors are required to measure c-di-GMP levels at different physiological regimes while keeping the focus of the biological sensor application in our manuscript.

On a more technical note, flow cytometry data are plotted as 'mean +/- SD', which only take peak values into account, while ignoring distributions. Population distributions (e.g., histograms of single cell data) should be shown directly in the main figures or in the supplementary or source data to allow the reader to judge how well their sensors separate populations with different c-di-GMP concentrations. In fact, dot plots in Fig. 1e,f give the impression that c-di-GMP high and low populations for both D3 and F8 biosensors largely overlap and are poorly separated. Similarly, the dose response curves in Fig. 2b and Extended Data Fig. 2 show large error bars even for mean values, which would imply that population distributions give an even noisier picture.

Response: We have made our data presentation clearer (see Fig.1b,c for FRET ratio distribution) and commented on that in the text. We would like to point out that it is normal that the FRET signal distributions in the populations of two different strain with very different median c-di-GMP levels overlap. For example, in PMID: 23163901, FRET ratio distributions also overlapped for strains with medium level and high level

of c-di-GMP. Given the large sample size, this overlap of distributions does not mean that the median values are not significantly different.

In Supplementary Fig.1d, we now showed histograms of fluorescence intensities for each channel (I_1 , I_2 and I_3) in flow cytometry data. In our study, we all used median values in each channel, to calculate FRET ratios or to calculate FRET efficiency. The error bars in the dose response curve have been reduced by testing more replicates.

Overall, the novelty of this work lies in the development of improved FRET sensors and not necessarily in the combination of FRET sensors with FACS as highlighted in the title (similar screens have been published before; PMID: 30395379, PMID: 26060330). Given that similar FRET-based sensors (PMID: 20522779, PMID: 23163901) and other biosensors for c-di-GMP (PMID: 34961989, PMID: 38724508) already exist, it is important to demonstrate that the new sensors are superior and offer added value to existing tools.

Response: We would argue that FRET-To-Sort is indeed a novel application, because it uses FRET-biosensor based fluorescence-activated cell sorting (FACS) of a barcoded transposon library for a genome-wide mapping of c-di-GMP regulatory network, which has not been reported before. Instead, in PMID: 30395379 and PMID: 26060330, compound screenings were respectively conducted by measuring FRET ratios using a plate reader and flow cytometry but not FACS. Our FRET-To-Sort application is performed to sort for mutations showing high or low c-di-GMP levels in a high-throughput way, which is a different screening. We now better highlighted these differences in the revised manuscript, and we appreciate that the Reviewer for pointing out the need to clarify it. Also see our response above regarding comparison to the already available real-time c-di-GMP biosensors.

Additional comments:

- Please provide single event data for all flow cytometry (and microscopy) measurements, e.g., as histograms in supplementary files.

Response: We have now displayed the example histograms of FRET ratios in Fig.1b,c, and example histograms of fluorescence intensities for each channel (I_1 , I_2 and I_3) of flow cytometry data in Supplementary Fig.1d. We have explained details in the calibration of bleed-through and calculation of FRET efficiency from cytometry data in Supplementary Fig.1a-d. Throughout the entire manuscript, we had thousands of measurements, and we cannot display all thousands of histograms in supplementary files. The original flow cytometry .fcs files containing single-cell data have been uploaded in the data sever.

For photobleaching microscopy data and flow FRET microscopy data, we did measurements at the population level, so no single-cell data are available in this case.

*- Please include statistics. E.g., Supp Fig 2: most biosensors do not seem to differ between *pdeH* and *dgcE* strains, but they state in the main text that this figure served to exclude biosensors that did not respond. Yet B1, F2, E2, C1 and F8, which show no difference in this Figure, were still included in Fig 2b and others.*

Response: We thank the Reviewer for this comment and sincerely apologize for the unclarity of presentation. This was caused by us combining the data from the FRET measurements with and without additional NaCl in the buffer in this figure. Since addition of NaCl reduced c-di-GMP levels in $\Delta pdeH$ strains, it lowered the difference between $\Delta pdeH$ and $\Delta dgcE$ strains for the low-affinity biosensors mentioned by the Reviewer. To improve clarity and also to streamline the focus of the manuscript (also see our response to the point below), we have now revised the manuscript to remove the measurements with NaCl added. Moreover, we also have included statistical analysis in this figure.

- The observation that ionic strength influences c-di-GMP levels is interesting and is certainly worth mentioning in the manuscript, but given that the underlying mechanism/cause is not addressed, does not seem to merit an entire paragraph.

Response: We agree with the Reviewer that this is an interesting but so far only preliminary observation. We have now removed the ionic strength part from our revised manuscript, to better focus it on the description and characterization of the FRET-biosensor toolbox and its novel application in combination with transposon mutagenesis and cell sorting (FRET-To-Sort), with additional follow-up analysis of the discovered dual effect of flagellar genes on c-di-GMP levels. Since our current manuscript has reached the limit, we decided to remove the ionic strength part.

- The genetic screen is only peripherally related to the main topic of the manuscript and fails to showcase the potential strengths of a FRET sensor. In fact, it is not clear why a FRET-based reporter would be required for such a screen at all. A simple transcriptional reporter would do the same trick. Also, the authors use both C3 (medium affinity) and D3 (high affinity) sensors in their screens (with different salt concentrations to achieve different basal c-di-GMP levels) and find 'similar results' (l. 203). Why did the authors perform the experiment in this way (different salt concentrations), rather than using the same conditions and see if they, in fact, get different hits depending on the KD of the sensor? This undermines the narrative that biosensors with different affinities are required for such studies.

Response: We thank the Reviewer for highlighting the necessity to better explain the advantages of the FRET biosensor toolbox for performing this screen, which we have now done in the revised version of the manuscript. Briefly, compared with transcriptional reporter, FRET-based biosensors have advantages of giving direct and immediate read-out of c-di-GMP levels, and they are not sensitive to the biosensor expression levels (which we now illustrate in the new Supplementary Fig.1g). In our experience, this is very critical for a reliable screening of mutants with high or low c-di-GMP levels. In mammalian cells, the integration of FRET biosensors and FACS sorting has been a highly successful approach (for instance, to identify key genes in the metabolic control (PMID: 30148842)), however, such combination has not been previously explored in bacteria. Furthermore, as illustrated in new Supplementary Fig.4b, biosensors with different affinities are required to sensitively detect differences in the c-di-GMP levels either below or above that in the wildtype, both in the initial screen and in the follow-up analysis of flagellar mutations, which we now better explain in the text. As mentioned above, the results related to varying ionic strength (which is indeed a specific effect in *E. coli*) are no longer included, and we only show the applications of the different-affinity biosensors as suggested by the Reviewer.

- One would have expected that in their genetic screen for c-di-GMP high mutants, the authors identify *pdeH* itself. Is there an explanation for why this was not the case?

Response: We agree with the Reviewer. As a matter of fact, the *pdeH* mutation was indeed strongly enriched in the second (x2) sorting cycle, but it had too few reads in one of the reference replicates in the third (x3) sorting cycle, likely due to an unspecific mutant loss due to repeated growth. We now show the data for gene enrichment for both sorting cycles (Tables 2 and 3). Since the selection for mutants with high c-di-GMP was highly efficient already after the x2 sorting cycle (Fig. 3b and Table 2), we now primarily relied on the data of the x2 sorting for our NGS analysis of mutants high c-di-GMP levels in the revised manuscript, to reduce the probability of potential unspecific mutant loss from the population during repeated growth.

Reviewer #2 (Remarks to the Author):

*Wang et al. have modified a previously published 92-member chemiluminescent YcgR biosensor toolkit (BRET, Reference 28), to a FRET-based one, because the former did not work as well in Gram negative bacteria. They characterized and selected a subset of these biosensors with different affinities for c-di-GMP, and used these to (1) test the response of the highly active DgcE to NaCl and 2) transform them into a random-barcoded Tn5 mutant library to identify mutants with low or high c-di-GMP determined by FACS sorting. While the specific experiments with DgcE conclude that its activity is responsive to ionic but not to non-ionic osmolytes, the data are not straightforward and raise more questions than they answer as noted below. The FRET-to-SORT part of this work is methodology development. It identified genes, which when mutated, show high or low c-di-GMP levels. The majority of hits that show increased c-di-GMP are in the well-studied flagellar regulon, which is not surprising because *pdeH* is a late flagellar gene and therefore will not be expressed in the absence of early flagellar genes. However, mutations that lower c-di-GMP map to flagellar genes *motA*, *motB* and *fliC* that are not expected to interfere with *pdeH* expression. Why this is so is not investigated. In other words, no significant new insight into the c-di-GMP network in *E. coli* has emerged from use of this toolbox.*

Response: We appreciate the Reviewer's feedback, which we have used to improve the clarity and focus of our manuscript. As noted by Reviewer #1, real-time biosensors are highly tools, and our toolbox of multiple biosensors with a consistent design but complementary and gradually differing affinities, including high-affinity biosensors, is a major advance compared to existing biosensors. We have now better highlighted the novelty and importance of our biosensor toolbox in our manuscript. FRET-To-Sort is indeed also a novel methodology development, which has not been reported before and is potentially broadly applicable in bacteria.

Regarding the biological implications of our findings, we have now stronger focused the manuscript on the observed impacts of flagellar gene mutations. Besides validating the proposed mechanism of mutations in flagellar class II genes, we have now investigated in depth the mechanism of downregulation of c-di-GMP levels by mutations in flagellar class III genes, which is highly surprising as noted by the Reviewer. Our data suggest that these effects are mediated by perturbations in the membrane potential sensed by the major cyclase DgcE. This novel mechanism has potentially general implications for the

physiological regulation of c-di-GMP in *E. coli*, including flagella-dependent mechanosensing. Besides flagellar genes, we discovered a number of other genes that affect c-di-GMP levels, including those related to cell envelope, fimbriae, stress response, and metabolism. Since most of the identified genes were not previously associated with the c-di-GMP regulatory networks, this broadens our understanding of c-di-GMP regulation and provides leads for future studies.

As also mentioned in our response to comments of Reviewer #1, we have now removed all data related to the effects of ionic osmolytes, to improve the focus of our revised manuscript. We decided to pursue the study of the ionic strength in a separate, independent manuscript.

Major comments related to DgcE being a sensor of ionic osmolytes

- 1. Fig. 3c presents c-di-GMP levels as measured by LC-MS/MS, which indicate that these levels are sensitive to NaCl in WT and DpdeH. The DdgcE mutant is unresponsive and has 5000 times less c-di-GMP compared to DpdeH, implying that DgcE is contributing to practically ALL the c-di-GMP in the cell. Missing in this analysis is important data for the DpdeHDdgcE double mutant, because subsequent experiments in Fig. 3e-g all use the double mutant (orange line) and show substantially high levels of c-di-GMP i.e. only 10% lower than WT. So, there appears to be a major discrepancy between what the biosensors measure and the actual concentrations of c-di-GMP.*
- 2. Fig. 3e,f. The authors claim that c-di-GMP levels are not affected by NaCl or Kglutamate in the DpdeHDdgcE mutant (orange line) compared to WT even though the data looks exactly like WT in which these levels ARE affected.*
- 3. The conclusion that DgcE is responsive to ionic osmolytes is premature in that the authors are monitoring c-di-GMP levels in the complete absence of this DGC in all their experiments. If DgcE associates with partners in the membrane, the effect could be indirect. It would have been more appropriate to use an active site mutant instead.*
- 4. The authors invoke depolarization of the membrane and perturbation of membrane potential to explain the effect of NaCl on DgcE activity, but no actual measurements are made for the only 'new' function they claim to have uncovered in this paper.*

Response: As mentioned above, we have decided to remove the ionic osmolytes part in the revised manuscript and rather refocused it on the biosensor library, FRET-To-Sort and follow-up investigation of effects of flagellar mutations. Nevertheless, we appreciate these comments that will be useful for our separate study of the ionic strength effect, and we have now used an active site mutant of DgcE as suggested by the Reviewer to investigate the impact of flagellar mutations on c-di-GMP levels.

Other comments

Although 18 sensors were selected for their sensitivity, only two are used in all the analyses without much explanation.

Response: We apologize for now clarifying it better. We have now included a more extensive explanation/justification of the use of different biosensors for the screen and follow-up analyses in the manuscript.

Line 38: While it is true that transcriptional sensors may be sensitive to other cellular factors, FRET-based approaches are not immune to such influences either.

Response: We have revised this part of the text. As we also mention in our response to the other reviewer, one key advantage is that our ratiometric FRET sensors are not sensitive to the sensor expression level (which we now illustrate in the new Supplementary Fig.1g). This is a crucial advantage for a reliable screening of mutants with high or low c-di-GMP levels, including those mutants that affect cell growth rate and thus generally levels of gene expression.

Lines 46-47: This recent publication suggests otherwise: <https://www.nature.com/articles/s41467-024-48295-0>

Response: We thank the Reviewer for this comment. The figure below shows the publication mentioned by the Reviewer. The yellow area shows the c-di-GMP range that can be sensitively readout (~0.4-4 μ M), which is indeed about a ten-fold change of c-di-GMP concentrations. We have now revised the statement to make it clearer.

Lines 78-81: The cited review does not propose that DgcE is a major DGC contributing to the global pool of c-di-GMP. Previous studies have shown that deleting DgcE does not significantly affect overall c-di-GMP levels, although it does influence biofilm production (PMID: 29018125). The authors may want to address and clarify the discrepancy between their findings and these earlier results.

Response: The cited review (PMID: 28163311) does state that in *E.coli*, “the global DgcE and PdeH module controls the overall levels of c-di-GMP”. In PMID: 29018125, they showed that in $\Delta pdeH$ backgrounds, *dgcE* mutations showed the largest decrease of c-di-GMP levels compared with other DGC. The figure in the same paper showing that DgcE does not recognizably affect overall c-di-GMP levels in wildtype backgrounds is likely simply due to the limit of detection in low c-di-GMP levels. Indeed, the same lab published a more recent paper (PMID:31022167), which demonstrated a significant decrease of c-di-GMP levels in $\Delta dgcE$ compared with wildtype cells.

Line 162: In Fig. 3B, why are the data plots for NaCl concentration at 60mM absent, when this is approximately the concentration used in tall heir buffers? In Extended Fig. 3 it is unclear what the different NaCl concentrations refer to.

Response: As mentioned above, we have decided to remove the NaCl part in the current manuscript.

Line 170: What is the rationale for switching the sensor A3 in Fig 3d to C3 in Fig. 3e-g? Based on previous data, both are medium-affinity biosensors.

Response: While we removed the NaCl part, we have revised the manuscript to better explain how we chose different biosensors.

Line 201: Fig. 3a shows no response of the D3 sensor to 67mM NaCl. Why then is 134mM NaCl used in sorting experiment with this sensor? Is it because it lowers c-di-GMP levels even further?

Response: We have removed the NaCl part in the current manuscript and now only use different-affinity sensors instead of changing ionic strength.

Line 209: If the low-affinity biosensor is more sensitive to decreases in c-di-GMP levels (line 167), why is the high affinity D3 sensor being used to screen for cells with low c-di-GMP levels as shown in Fig. 4a?

Response: We thank the Reviewer for pointing out this unclarity. The rational for using different-affinity biosensors is now better clarified in the manuscript and illustrated in Supplementary Fig.4b.

Line 230: BluF is involved in biofilm maturation, but it is not related to c-di-GMP production; BluF has a degenerate EAL domain.

Response: We apologize for the misleading statement. We updated the text accordingly.

Line 243: Extended Fig. 7a. If these transposon mutants were selected based on FRET signal higher than pdeH, why are all the mutants showing a lower signal compared to pdeH?

Response: We apologize for this partly incorrect statement. We were at first indeed selecting only for mutations with the FRET signal above that in *pdeH*, but subsequently relaxed the selection to choose mutants showing higher or comparable FRET efficiency as in $\Delta pdeH$ (higher than 80% of the FRET efficiency value in $\Delta pdeH$). That is why we identify candidate mutations of, for instance, *fliG* and *fliH*, showing slightly lower FRET efficiency than $\Delta pdeH$ but much higher than wildtype cells. We have revised the text accordingly.

Reviewer #3 (Remarks to the Author):

Reviewer #4 (Remarks to the Author):

Response: We also thank these two Reviewers for their very helpful feedback on our manuscript.

REVIEWER COMMENTS

Reviewer #4 (Remarks to the Author):

Summary:

The manuscript by Wang et al. reports the development of a FRET-based biosensor kit capable of detecting a broad range of c-di-GMP concentrations in real time, spanning from 9 nM to 1 μM. The study screened 16 distinct sensor variants, each exhibiting different affinities for c-di-GMP, thereby enabling researchers to select sensors best suited for specific experimental conditions or bacterial species of interest. Using a microfluidic platform, the authors monitored live cells harboring the biosensors and observed remarkable cell-to-cell heterogeneity in intracellular c-di-GMP levels. They further integrated the FRET-based sensors with fluorescence-activated cell sorting (FACS) to perform high-throughput screening of a E. coli Tn5 transposon mutant library, identifying mutants with either elevated or reduced c-di-GMP levels relative to ΔdgcE and ΔpdeH reference strains. Interestingly, the screening revealed opposing effects of mutations in flagellar class II versus class III genes: disruptions in class II genes led to an increase in c-di-GMP levels, whereas class III mutations caused a decrease. The authors further linked the reduction in c-di-GMP levels to an increase in membrane potential, suggesting a mechanistic connection between flagellar assembly defects and c-di-GMP signaling dynamics.

The authors have addressed most of the concerns raised in the previous version of the manuscript and provide new insights into the potential relationship between membrane potential and c-di-GMP dynamics in E. coli. The manuscript could be further strengthened and clarified by addressing the comments outlined below.

Previous comments:

- 1. All comments regarding osmolytes are no longer relevant, as the authors have decided to remove this section entirely and focus instead on validating the sensor and explaining changes in c-di-GMP levels resulting from mutations in flagellar class II or III genes.*
- 2. The justification for using different sensors in each assay has been provided and is now easier to understand.*
- 3. Minor textual revisions have been made, improving overall clarity.*

Response: We thank the Reviewer for the positive evaluation of our revised manuscript, and appreciate the acknowledgment that most of the previous questions have been satisfactorily addressed. We are also grateful for additional comments, which have helped us to further improve the presentation of our data. We have revised the manuscript to address all of the comments, as outlined below.

Comments on the Revised Manuscript:

1. In line 107, the authors previously stated that FRET efficiency provides a more accurate readout than the FRET ratio, as it is less influenced by bleed-through and biosensor expression levels, which makes the FRET ratio less suitable for live-cell tracking. However, in Supplementary Figure 7 (Figure S7), the authors use the FRET ratio to measure c-di-GMP levels in live cells. While this choice is understandable given potential technical limitations, the earlier emphasis on FRET efficiency as the better metric raises concerns about the robustness of the assay.

Response: We thank the Reviewer for raising this question, which we agree requires better clarification. Our choice of readout metric depends on the experimental context. In flow cytometry experiments comparing different biosensors and different strains, biosensor expression levels can vary substantially among samples. Since, as shown in Supplementary Fig. 1g,h, FRET efficiency is less sensitive to expression variability because bleed-through is explicitly calibrated, it proved to be a more reliable FRET measure for systematic quantification in flow cytometry measurements.

In our microscopy experiments shown in Supplementary Fig. 7, the comparisons were instead performed for single cells of the same strain, and in this case the FRET ratio was not apparently affected by the limited expression variability (shown in new Supplementary Fig. 7b). Moreover, here we did not aim to directly compare FRET efficiencies between different biosensors, only their sensitivity to c-di-GMP levels. Although the calculation of FRET efficiency instead of the FRET ratio is indeed possible in microscopy experiments, too, using the so-called “three-cube” FRET, this would complicate the microscopy setup, reduce time resolution and increase photodamage. Therefore, in these experiments, the FRET ratio serves as a simple and reliable proxy for changes in intracellular c-di-GMP levels in individual cells. It was already used in our previous work (we now added the corresponding citation). We have now better explain the rationale for our use of the FRET efficiency for flow cytometric measurement, and the FRET ratio for live-cell microscopy in the revised manuscript. Please see Supplementary Fig. 7 and Lines 106-113, 117-119 and 177-179.

2. The authors use Figure S7 to support their conclusion that the sensor can monitor c-di-GMP levels in live cells and that “high dynamics and heterogeneity of c-di-GMP levels in growing *E. coli* cells, which were not apparently correlated with the cell cycle” (line 167). However, when comparing the performance of the F1 sensor under two conditions—Figure 2d and Figure S7a—it remains unclear:

a) whether comparable output levels should be expected for equivalent c-di-GMP concentrations, and
b) if so, whether the data in Figure S7a imply that intracellular c-di-GMP levels among individual cells range from approximately 10 nM to 810 nM.

Although prior studies have reported heterogeneity in c-di-GMP concentrations, it is uncertain whether the present data substantiate such a wide dynamic range. It would strengthen the manuscript if the authors could further discuss the expected magnitude of these dynamics based on their data and how their observations compare with previously reported ranges.

Response: We thank the Reviewer for this comment. The values of the FRET ratio in Supplementary Fig. 7a and in Fig. 2d are not directly comparable, because in the former we measured FRET signals in intact cells whereas the latter measurements were done in permeabilized cells. This difference is known to affect properties of both fluorescent proteins (e.g., due to effects of salts, pH etc *in vitro* measurements), and thus the absolute value to the FRET ratio. Moreover, as mentioned in the Methods, the two experiments were performed on different microscopy setups.

Nevertheless, we expect the sensitivity range of the F1 sensor to be the same or similar in the intact cells in microfluidics experiments as in the permeabilized cells (Fig. 2b), ~10-100 nM. As we discuss in the revised manuscript, our data do suggest that measured wildtype single-cell levels cover this entire range. We now show that the FRET ratio values in individual wildtype cells vary from those similar to those in the $\Delta dgcE$ strain (mean FRET ratio ~1.4) to those similar to the $\Delta pdeH$ strain (mean FRET ratio ~ 2.0; in this latter case the c-di-GMP levels are expected to be ~630 nM (PMID: 29018125) and likely saturate the sensor. However, smaller variability measured using the low-affinity C1 biosensor (K_D ~370 nM) suggests that, the levels of c-di-GMP in wildtype cells remain in most cells below the sensitivity range of this sensor (i.e., below ~100 nM). We have revised the manuscript to accordingly discuss the expected magnitude of c-di-GMP dynamics. Please see Supplementary Fig. 7 and Lines 179-186.

3. In both Figure 2b and Supplementary Figure 4a, the 16 FRET sensors exhibit the dose-dependent responses described by the authors. While most sensors show relatively low variability within the concentration range suitable for measuring c-di-GMP levels in wild-type *E. coli* cells, the substantial standard deviations observed at higher concentrations (approximately 100–1000 nM) make it difficult to determine whether most of these sensors are also reliable for detecting elevated c-di-GMP levels in bacteria such as *P. aeruginosa*.

Response: We apologize for potentially misleading representation. The apparent variability at higher ligand concentrations is simply due to the technical variability in our measurements, as in the previous version of the manuscript we were showing standard deviations (SD). To improve clarity, we have revised Fig. 2b and Supplementary Fig. 4a to instead display the standard error of the mean (SEM), which better represents the precision of the measurement in this case.

4. In line 286 and Figure 4D–E, DNP was used to dissipate the proton motive force (PMF) in wild-type cells, resulting in an observed increase in c-di-GMP levels. It is unclear, however, whether the same treatment was applied to the *motA* and *fliC* deletion mutants. Based on Figure 4D, where DNP collapses most of the PMF, one would expect these mutants to exhibit c-di-GMP levels comparable to those of the DNP-treated wild type, despite their inherently elevated membrane potential. Including this control would strengthen the interpretation that the increased membrane potential in these mutants contributes to the observed decrease in c-di-GMP levels.

Response: We thank the Reviewer for this valuable suggestion. We have now included the data for Δ *motA* and Δ *fliC* mutants treated with DNP in Fig. 4d-f. As expected by the Reviewer (and by us), in both mutants DNP treatment largely collapsed the PMF, also it remained slightly higher than that of DNP-treated wildtype cells. This nicely matches the c-di-GMP values, with these DNP-treated mutants showed increased c-di-GMP values, but slightly lower than c-di-GMP levels in DNP-treated wildtype cells.

5. In line 299, are the authors suggesting that c-di-GMP levels decrease upon surface contact as a consequence of increased membrane potential? It would be important to clarify whether the antibody concentration used in the experiment fully inhibits motility, as this detail is crucial for interpreting the proposed mechanism. The prevailing understanding is that c-di-GMP levels generally rise upon surface contact, although the timing and extent of this increase can vary. As mentioned in line 281, stator engagement is load-dependent; however, it remains unclear whether antibody binding is perceived by the cells as an increased mechanical load, potentially triggering additional stator recruitment. Moreover, do the authors observe a similar decrease in c-di-GMP levels when cells experience increasing gradients of viscosity? Please clarify lines 300–302, which currently suggest that surface contact leads to reduced c-di-GMP production, or expand the discussion to reconcile this observation with existing models.

Response: We thank the Reviewer for these insightful points. Firstly, to confirm that the antibody concentration used in the experiment fully inhibited motility, we have now included analysis of the swimming motility of wildtype cells with and without antibody treatment (new Supplementary Movies 1-2, and new Supplementary Fig. 12). While we agree with the Reviewer that stator engagement is load-dependent, the antibody binding under our conditions completely inhibits motility, meaning that there is no motor rotation - and thus no dissipation of PMF through the motor – even with these additionally recruited stators, explaining our results. As suggested by the Reviewer, we have now further added the

data showing that an increase of environmental viscosity using 2% ficoll resulted both in slower swimming and in a decrease in the FRET ratio, indicating lower c-di-GMP levels (new Supplementary Fig. 13). Thus, these data further support the antibody results.

We also agree with the Reviewer that our results contradict the prevalent model where bacteria are typically expected to increase c-di-GMP upon immobilization of flagellar during surface attachment (although the interplay between inhibition of motility and c-di-GMP regulation is frequently more complex, as elaborated in our very recent review paper on this topic). This deviation was already briefly mentioned in the Discussion in the previous version of the manuscript. We now additionally performed c-di-GMP level measurements under conditions of early static biofilm formation by *E. coli*, comparing FRET in cells attached on the surface and cells in floating aggregates above the surface using confocal microscopy. In agreement with our antibody treatment and ficoll data, these results indeed showed that cells immobilized on the surface exhibit significantly lower c-di-GMP levels compared to cells in aggregates above. These findings are now shown in new Supplementary Fig. 14. We have further expanded general discussion of the deviation of our observations from the previously established paradigm. Please see Lines 408-417 and 418-423.

Minor comment

1. Given that the toolkit contains 16 sensors, what would be the most straightforward approach to determine which sensor is best suited for different bacterial species or experimental conditions? Can the dissociation constants and FRET efficiencies characterized in *E. coli* across various c-di-GMP concentrations serve as an initial reference point, or would users need to conduct screening experiments similar to those performed by the authors to identify the optimal sensor? As this study introduces a new experimental tool, it would be helpful—though not strictly necessary—to include practical guidance or recommendations for sensor selection.

Response: Yes, since our sensors function autonomously and do not rely on any specific *E. coli* proteins (and show similar performance *in vivo* and *in vitro*), we believe that they are universally applicable – assuming that they can be expressed in the target organism and the c-di-GMP levels are within the range of our toolbox. We now mention it in the Discussion.

Reviewer #5 (Remarks to the Author):

In the manuscript “A toolbox of FRET-based c-di-GMP biosensors and its FRET-To-Sort application for genome-wide mapping of the second messenger regulatory network,” Wang et al describe the production of a set of FRET biosensors for the secondary messenger cyclic-di-GMP and its use through a FACS-based transposon mutagenesis screen. This screen identified two sets of flagellar mutants that affected c-di-GMP levels: one set which increased c-di-GMP in a previously identified PdeH-based mechanism, and a second set that decreased c-di-GMP. The authors investigated the latter and found some evidence to suggest that this was related to disruption of membrane potential through an inactivation of the flagellar machinery. I was asked to serve as a secondary reviewer for this manuscript as the initial Reviewer #1 was unable to review this resubmitted version. I am therefore primarily focusing on the response to the previous reviews and the new material regarding membrane potential regulating c-di-GMP concentrations in flagellar mutants, as I have not read or reviewed the prior manuscript submission containing the NaCl data.

In regards to the response to the prior reviews, the authors have done a good job clarifying where biosensors are being used, and their purification and in vitro testing of the biosensor Kds alongside the permeabilization assay has supported their prior findings, which are both positive. However, there are still several concerns with these responses and with the new data on membrane potential regulation of c-di-GMP:

Response: We thank the Reviewer for acknowledging the improvements made in the previous revised revision. We also appreciate the Reviewer highlighting the following remaining points of concern, which we have carefully considered and addressed, as described below.

- The authors mention that one novelty of this work is the production of a biosensor library with a range of binding affinities. In PMID: 23163901 mentioned by the prior first reviewer, a biosensor library consisting of Kds ranging from 88nM to 8.24uM were produced, covering roughly a range of 27nM to 26uM using their one-log coverage range. So while the authors are correct that their especially high affinity biosensors are new to the biosensor field, the concept of using a range of binding affinities is established.*

Response: The concept of establishing a range of affinities for c-di-GMP biosensors has indeed been mentioned before. But while those binding affinities reported in PMID: 23163901 covered a wide range, the number of biosensors was limited. Since each individual sensor can only sensitively respond to a narrow range of c-di-GMP levels, we argue that investigation of cellular c-di-GMP concentrations under diverse physiological conditions will benefit from our larger set of sensors with a consistent readout and a stepwise coverage of binding affinities. As stated in the manuscript, it is the combination of large range of affinities, including high affinities (as acknowledged by the Reviewer), and the stepwise coverage of affinities within this range that makes our toolbox superior.

- While there are other publications that use flow cytometry to measure biosensors, I agree that incorporating a sorting and Tn-Seq based analysis is a useful tool for further study. Since the sorting is the novel component though, I'd like to see a little more data on how effectively mutants were sorted. Higher c-di-GMP bacteria especially may be more prone to clumping together (PMID: 26060330), which could cause a subsequent mutant that was similarly sticky to be brought along during sorting. From my reading, it sounds like the authors sorted for c-di-GMP and subjected a partial mixture to individual colony isolation. Clones that showed either high or low c-di-GMP were Sanger sequenced, but how efficient was isolation? In other words, how many sorted/isolated colonies no longer showed altered c-di-GMP that were excluded from further testing, but that might appear in the NGS sample? Along the same lines, the high c-di-GMP batch appeared to be moderately efficient, with about a third of the NGS-derived clones also being isolated through Sanger sequencing, but there were also clones that were only isolated through Sanger or NGS but not both. This was much more pronounced with the low c-di-GMP set, in which only 10% of indicated mutants were isolated through Sanger sequencing. Is this just a case of a small sample size in Sanger failing to quantify the full mutant set? Or is the sorting accidentally pulling out mutants that appear in NGS data but that were excluded from Sanger sequencing because their c-di-GMP levels don't actually change? Why also is the Sanger sequencing identifying mutants that weren't identified in the NGS sample? Are these Sanger-identified mutants closer to WT c-di-GMP to the point where they didn't make the NGS cutoff? Considering this low data set is their novel high affinity biosensor and the sorting is the novel component for this technique, the authors should comment on these apparent discrepancies and limitations.*

Response: We thank the Reviewer for acknowledging the novelty and utility of the FRET-To-Sort approach and for raising points regarding this procedure that may require further clarification. We apologize that several aspects of our screen have not been described in sufficient detail in the previous version of the manuscript.

Regarding the concern that high c-di-GMP mutants may aggregate during sorting: we grew *E. coli* cultures, including transposon library for sorting, at 37°C, a condition known to suppress curli matrix expression and thus to reduce aggregation in *E. coli* K-12 strains. While some aggregation – mediated by an autoadhesin, antigen 43, can still be observed even under these conditions, these aggregates are highly labile and easily disrupted by vortexing, which we routinely perform before cytometry/sorting (this is now explicitly mentioned in the Methods; we also refer to our previous publications where this was shown to ensure aggregate disruption). Consistent with this, we did not observe aggregation either before or during sorting, as illustrated in the forward scatter (FSC) plots below:

Figure: FSC plots of wildtype cells and Tn5 transposon library, incubated and measured on the same day.

As for an only partial overlap between the Sanger and NGS data, one reason is that the NGS samples underwent an additional overnight growth step to obtain sufficient material for sequencing, whereas the Sanger-sequenced clones did not. As a result, mutants with reduced growth may be underrepresented or absent in the NGS dataset but still appear in Sanger sequencing. Even more importantly, only a relatively small number of colonies were Sanger sequenced as a reference, which led to many candidates appearing only in NGS data but not in Sanger data. These points are now better explained in the corresponding Results section (Lines 217-224).

• In Figure 4, the authors investigate the role of class II and class III flagellar mutations on c-di-GMP. The class II mutant model makes sense and is well supported by prior work, where a failure to secrete FlgM resulting in low expression of PdeH and high c-di-GMP. The authors hypothesize that the low levels of c-di-GMP seen in class III mutants may be due to over-secretion of FlgM, but test that using a flgM mutant. Wouldn't the flgM mutant just mimic FlgM over-secretion? For instance, if the fliC mutation is causing a higher than typical FlgM secretion and deleting flgM completely lowers c-di-GMP even further, it would seem that would provide evidence to the FlgM over-expression hypothesis. I don't think these experiments show a FlgM-independent effect for the type III mutants as the authors claim, at least based on this data.

Response: We apologize for not explaining the rationale and the interpretation of these experiments more clearly, and thank the Reviewer for drawing our attention to the need to do that. In brief, the fact that the effect of class III (*fliC* and *motA*) mutations on c-di-GMP levels is observed even in the absence of FlgM in the background (i.e., when comparing *flgM fliC* or *flgM motA* mutants to *flgM* strain) means that the effect of these mutations is not FlgM-dependent. This is in contrast to FlgM-dependent effects of class II mutations, where their effect on c-di-GMP disappears in the absence of *flgM* (or, more precisely, even reverses to resemble the effect of class III mutations). We now better explain it in the corresponding Results section (Lines 280-282).

• *In Fig. 4c, the authors investigate this further, but have to use a pdeH mutant in order to move c-di-GMP levels into a range detectable by their biosensors, claiming that the further reduction of c-di-GMP beyond the fliC mutant level may not be possible with their high affinity (D3) biosensor. Were these experiments attempted with the D3 biosensor? Considering these high affinity biosensors are the one novel set of biosensors, this would seem to be an excellent test case for their utility. One potential downside to measuring these effects at higher concentrations is that you may be reaching the DGC I-site binding affinity for DgcE, causing it to repress independent of the desired response. Is there a predicted level of c-di-GMP at these various biosensor levels, and is the I-site Kd for DgcE known?*

Response: We thank the Reviewer for the suggestion to show these data. Indeed, these experiments in Fig. 4c had been attempted with the D3 biosensor, as now shown in new Supplementary Fig. 9. As mentioned, due to its high affinity, D3 biosensor saturated in $\Delta pdeH$ background strains, because of its relative high c-di-GMP levels, and thus it was not suitable for quantifying differences as done with the C3 sensor in Fig. 4c.

Regarding potential inhibitory c-di-GMP binding, the I-site dissociation constant (Kd) for DgcE is, to our knowledge, not known. However, since the impact of the mutations of *motA* and *fliC* is to reduce c-di-GMP levels, such potential autoinhibition via the I-site should not confound the interpretation of our result in Fig.4c.

• *In Figure 4g, the authors see a small uptick in FRET efficiency upon treatment with DNP in a DgcE-dependent manner. However, they're using a low affinity F2 biosensor in this figure that is displaying a FRET efficiency of around 24%. In Supplementary Figure 3, their F2 biosensor appears to bottom out in the dgcE mutant at 30%, ranging from 30-45%. Are these FRET efficiencies calculated differently between figures? Is it possible that the biosensor is unable to detect c-di-GMP concentrations below the point at which the authors are measuring? There's no low c-di-GMP control to show that these unchanged values aren't due to the fact that they are just outside the range of biosensor measurement. As a large point of the manuscript is the utility of these biosensors to detect ranges accurately, low and high c-di-GMP mutants (*pdeH* and *dgcE*) should be included in every figure to ensure the measurements are within that particular biosensor's range.*

Response: We thank the Reviewer for pointing these out. Because FRET efficiencies measured in Supplementary Fig.3 were performed several years ago during our initial screening and the cytometry equipment was relocated and underwent laser realignment and recalibration in the meantime, the baseline FRET efficiency values become apparently shifted while leaving the relative dynamic range of

a biosensor unchanged. We sincerely apologize for not clarifying this in the previous version of the manuscript, and we now mention this in the legend of Supplementary Fig.3. However, all measurements in Fig. 4 and corresponding SI figures were performed in the same setup and are self-consistent.

We fully agree with the Reviewer that a control of low c-di-GMP strain would make sure that the measurement of c-di-GMP levels in double mutations are not out of the detecting range of the biosensor. To address this, we have now included results of DNP-treated double mutations using the medium-affinity biosensor C3, which has been confirmed to be capable of detecting both up- and downregulation of c-di-GMP levels around the baseline in the strains. Please see the updated Fig. 4g, and Fig. 4c for *pdeH* and *dgcE* controls.

• *In Fig 4h, the authors use a flagellar antibody to attempt to halt rotation of the flagellum in an effort to indicate this alters membrane potential to inactivate DgcE. Have the authors confirmed these wildtype bacteria are motile at this concentration? According to Fig. 4a, the C3 biosensor appears to max out at a FRET efficiency of 37%, pretty similar to what is seen in the wildtype sample here. The C3 biosensor would apparently measure up to around 230nm, which at the higher end would be sufficient to activate the YcgR flagellar brake (Kd 141nM - PMID: 31836667). How does YcgR fit into this model? Wouldn't YcgR inhibition of flagellar rotation also then cause increased membrane polarization, inactivation of DgcE, reduction of c-di-GMP through PdeH, and subsequent release of the flagellar body by YcgR? This would seem to make YcgR fairly useless as a flagellar brake. The authors also mention that this antibody-based flagellar jamming could mimic surface contact, but don't describe why Salmonella would want to lower c-di-GMP upon surface contact. Wouldn't this be better served by high c-di-GMP and biofilm formation?*

Response: We thank the Reviewer for these comments. First, the wildtype cells used for antibody treatment were washed by centrifugation and then resuspended in the buffer prior to flow cytometry measurements, rather than directly diluting the day culture in the buffer as used for other measurements. This treatment reproducibly led to higher FRET levels in Fig. 4h compared to those for the wildtype with the same C3 biosensor in Fig. 4a or 4e. Although reasons why FRET/c-di-GMP increased upon this treatment were not further investigated here, this washing step consistently increased c-di-GMP levels in both wildtype cells and *motA* cells (compare Fig. 4h to Fig. 4a and 4e). We now mention this in the legend of Fig. 4, and apologize that it was not clarified previously.

We now show control experiments that confirm that wildtype bacteria are motile before the antibody treatment and ceased swimming after the treatment (Supplementary Movie 1-2, Supplementary Fig.12).

Additionally, to confirm our results obtained using the antibody-mediated inhibition of flagellar rotation, we now added further experiments that demonstrate similar effect upon increased medium viscosity (which similarly reduces flagellar rotation; new Supplementary Fig. 13) and also when comparing surface-attached cells with those in floating above the surface (new Supplementary Fig. 14). We have correspondingly expanded the discussion of these results, emphasizing that the observed reduction of c-di-GMP in cells with inhibited motor rotation, including surface-attached cells, indeed differs from the prevalent model where such inhibition should rather lead to increased c-di-GMP levels. But we feel that more work will be required to understand the physiological significance of this regulation, and we would rather not speculate about that in the current manuscript.

Regarding the role of YcgR, the Reviewer is absolutely right to point out this as an interesting question. While addressing it goes beyond this manuscript, we are currently investigating the impact of the observed motor-dependent c-di-GMP regulation on the YcgR function as a follow-up project. We do see that YcgR is functional in regulating the motor speed and thereby reducing c-di-GMP under our conditions. Our hypothesis that we are currently testing in the follow up work is thus that the YcgR/potential-mediated regulation might function as homeostatic feedback to control motor rotation speed.

- *The authors mention (line 337-338) that “perturbation of the cell envelope and membrane composition might generally affect c-di-GMP signaling.” They reference this in regard to both low and high mutants, but in their model wouldn’t these perturbations universally lead to lower c-di-GMP due to membrane polarization-induced DgcE inactivation? Without a mechanism, my assumption was that high membrane potential somehow inactivated DgcE that would normally be synthesizing c-di-GMP. Or is the suggestion that DgcE is specifically also activated by low membrane potential beyond wild type potential levels? That would be an interesting finding, but doesn’t appear to be discussed here. Absent a further biochemical examination of DgcE, the indication that it somehow responds to membrane potential needs better evidence.*

Response: The statement was intended to refer to cell envelope and membrane perturbations as a broad class of environmental inputs, which may lead to either increases or decreases in intracellular c-di-GMP levels, dependent of the perturbation type. These effects might, but do not need to, be mediated by changes in membrane potential specifically. Regarding another Reviewer’s question: Our results indeed suggest that DgcE activity is not only downregulated by the increased membrane potential (in *fliC* and *motA* mutants) but also upregulated by low membrane potential (illustrated by the DNP treatment; Fig. 4e). We now explicitly mention this in the Discussion. Please see Lines 403-407. The mechanism of this dependence of DgcE activity on membrane potential is the subject of our ongoing investigation.

- *The other issue with the author’s model is during class II flagellar mutant analysis. The beginning of the manuscript details how DgcE is the primary cyclase present in E. coli, and that reduction of PdeH levels results in DgcE-mediated c-di-GMP concentrations increasing. According to their model though, these class II mutations should also be disabling DgcE via increased membrane potential. In this situation, what cyclase is producing c-di-GMP in the class II mutants when both PdeH and DgcE are absent? If they test a fliG/dgcE double mutant, are c-di-GMP similarly as high as the fliG mutant? Or would these levels decrease because DgcE is still active despite the higher membrane potential? None of these double mutants were tested, and the class II mutants weren’t tested for membrane potential to provide support for their DgcE membrane potential sensor activity.*

Response: We thank the Reviewer for raising these important points for the clarification of our model. We have revised our manuscript regarding our model to clarify the logic of our model. In strains lacking both PdeH and DgcE, residual c-di-GMP production can be attributed to other diguanylate cyclases encoded by *E. coli*, as *E. coli* cells have multiple diguanylate cyclases in addition to the major cyclase DgcE, which could contribute to cellular c-di-GMP levels (PMID: 29018125).

Regarding the impact of class II mutants on the membrane potential, we now show these results in Supplementary Fig. 11. The class II flagellar mutations indeed lead to a significantly increased membrane

potential. In $\Delta flgM$ background strains, and thus in the absence of the dominant FlgM/PdeH dependent impact of class II mutations, such increase in the membrane potential does lower c-di-GMP levels (Fig. 4a), as expected from DgcE inhibition. We have revised the text to stronger emphasize how these results support our model.

Finally, the levels of c-di-GMP in class III (*fliC* or *motA*), or in class II mutants introduced in $\Delta pdeH$ background, are indeed slightly but significantly higher than those in the *dgcE* knockout or $\Delta pdeH \Delta dgcE$, respectively (Fig. 4b,c), confirming that the increased membrane potential in these class III and class II mutants largely but not completely inhibits DgcE. We now comment on this in the manuscript (Lines 317-320). Consistent with that, the *fliG dgcE* mutant does have lower c-di-GMP level than the *fliG* deletion alone (see figure below):

Figure: Impact of deletions of flagellar class II genes on c-di-GMP levels. Higher FRET efficiency indicates higher c-di-GMP levels. *P* values were calculated using unpaired two-tailed *t*-test, *n* = 3. ****P* < 0.001; ***P* < 0.01. Data are presented as mean \pm SD.

In summary, I think there are some useful tools in here. The high affinity biosensors could provide new avenues of study, although it is a little worrying the authors didn't capitalize on this when it was potentially useful. The FRET-to-SORT is also a potentially powerful tool with a little more data to indicate the efficiency of sorting and discrepancies between the Sanger and NGS datasets. The model of membrane polarization affecting DgcE and c-di-GMP levels is the truly novel part here, and there the work is still pretty preliminary. Inclusion of some more controls to support their model would be necessary, and potentially some biochemical mechanisms underlying DgcE detection of membrane potential. I don't know that the biosensor/sorting techniques are novel enough on their own without the membrane potential finding, but strengthening this latter set of findings would greatly improve the manuscript.

Response: We thank the Reviewer for this recognition of the utility of the high-affinity biosensors and the potential of the FRET-To-Sort approach, and for helpful suggestion on further improvement of our manuscript. We have now addressed these points, by expanding (and better explaining) the use of different-affinity biosensors and explaining the discrepancies between the Sanger and NGS enrichment

data in the revised manuscript. We also agree that the discovered motor rotation- / membrane potential-dependent regulation of DgcE is highly interesting and we have also added suggested control experiments and extended the discussion of this observation.